# A Fast Similarity Matrix Calibration Method with Incomplete Query

## ABSTRACT

The similarity matrix is at the core of similarity search problems. However, incomplete observations are ubiquitous in real scenarios making the similarity matrix less accurate. To estimate a high-quality similarity matrix, one popular trend is to impute the missing values into the vectors directly, which provides a simple and highly efficient way to recover the similarity matrix. However, these methods lack of theoretical guarantee due to ignoring the entire similarity matrix property directly. In this paper, based on the key insight that the similarity matrix is symmetric and enjoys the positive semi-definiteness (PSD) property, we proposed a novel similarity matrix calibration method, which is scalable, adaptive, and sound. Specifically, we first show the similarity matrix provably holds the PSD property as the constraint. Then, we proposed a parallel matrix calibration method to estimate the similarity matrix to approximate the unknown fully observed ground-truth similarity matrix. Further, we discover its factored form which bypasses the computation of singular values and allows fast optimization by general optimization algorithm. Stable recovery and convergence are guaranteed. Extensive similarity matrix calibration experiments on the real-world dataset demonstrated that the proposed method obtains superior performance while being the fastest in comparison to baseline methods.

## CCS CONCEPTS

• **Computing methodologies** → Machine learning methodes.

## KEYWORDS

Similarity Search, Incomplete Query, Positive Semi-definiteness, Similarity Matrix, Matrix Calibration

**ACM Reference Format:**
Paper ID: 791. 2023. A Fast Similarity Matrix Calibration Method with Incomplete Query. In *Proceedings of The Web Conference 2024*. ACM, New York, NY, USA, 13 pages. https://doi.org/XXXXXXX.XXXXXXX

## 1 INTRODUCTION

Similarity search with incomplete data has attracted extensive research attention in several research fields [4, 12, 13, 54]. However, data missing is ubiquitous and unavoidable due to various practical issues. In real scenarios, data missing is most likely caused by missing values directly in the data samples with certain features unknown, resulting in the similarity matrix being measured by the incomplete data samples. The other scenario is that the data missing occurs within the similarity matrix due to storage or transmission error.

To solve the data incompleteness problem and approximate the unknown complete data vectors, the Missing Value Imputation

*The Web Conference 2024, MAY 13 - 17, 2024, Singapore*
2023. ACM ISBN 978-1-4503-XXXX-X/18/06...$15.00
https://doi.org/XXXXXXX.XXXXXXX

(MVI) method [14, 39, 40] is proposed to impute the missing values directly without any data assumptions, including pair-wise deletion, mean substitution, regression, or the expectation-maximization algorithm. Besides, a series of algorithms are proposed with a detailed exploration of particular data samples [19, 31], which explores the internal property of various matrices and achieved the matrix competition on the high-rank matrix and low-rank matrix. However, the performance highly depends on the assumption of data samples and varies significantly to the data distribution. Moreover, the requirement of fast response time and high throughput of online scenarios often makes it impractical to apply the computational-aware MVI methods. On the contrary, the Matrix Calibration (MC) method [42, 53] imposes exact constraints on a specific matrix given based on specific properties. A series of studies on the similarity matrix imputation focus on the positive definiteness property [6, 37, 45], which is proved to be effective but is not efficient.

In this paper, we proposed a fundamentally different matrix calibration method and defined it as a convex optimization problem to find the feasible solution as the estimated similarity matrix and further apply it to solve the similarity search problem. Instead of imputing the missing observations directly, we started with an initial similarity matrix calculated by incomplete data samples and then modified the initial matrix to satisfy the certain constraint, positive semi-definiteness (PSD) [5], which is the exact property that the unknown ground truth similarity matrix should hold. The theoretical analysis showed that the well-estimated similarity matrix is guaranteed to be closer to the unknown ground truth. The empirical evaluations reported the superior performance of similarity matrix calibration and further verified the promising potential to solve the similarity search problem with incomplete observations.

In sum, we proposed a novel matrix calibration method and provided an efficient algorithm to tackle the similarity search problem with incomplete observations. Our contributions can be outlined as follows:

- **Theoretical Novelty**: We proposed a fundamentally different matrix calibration method to estimate the similarity matrix from the incomplete data and then modify this estimate to satisfy the positive semi-definiteness (PSD) property under the well-defined convex optimization problem. The proposed method provided theoretically guaranteed improvement to the unknown ground truth similarity matrix that was calculated by complete data samples.
- **Methodology Soundness**: We designed a convex optimization problem to minimize the difference between the estimated similarity matrix and the unknown ground truth. To achieve an efficient solution, we further simplified the optimization problem for similarity vectors with similar constraints, and then proposed scalable approximate algorithm.
- **Empirical Verification**: We conducted a series of experiments to verify the performance of the proposed method

to estimate the similarity matrix and further tackle the incomplete similarity search task. The evaluation involved the effectiveness, efficiency, and sensitivity. Our main method showed a clear improvement to estimate the similarity matrix. which provided a promising practical tool in similarity search applications.

**Notation**. Vectors are denoted in lowercase and matrices in uppercase. $(.)^\top$ denotes transpose operation. $I$ denotes the identity matrix. For a square matrix $X \in \mathbb{R}^{n \times n}$, $\|X\|_F$ denotes its Frobenius norm. Let the singular value decomposition (SVD) of matrix $X$ be $VSV^\top$, where $V \in \mathbb{R}^{n \times n}$, $S = diag(\sigma(X)) = [\sigma_i(X)]$ with $\sigma_i(X)$ being the $i$th singular value of $X$ and w.l.o.g., $\sigma_1(X) \geq \sigma_2(X) \geq ... \geq \sigma_n(X) \geq 0$.

## 2 BACKGROUND

### 2.1 Similarity Matrix

*2.1.1 Similarity Matrix Calculation.* Similarity matrix [41] measures the pairwise similarities between the data samples and lies on the positive definiteness property [53]. One of the widely used similarity metrics is Cosine Similarity [25]:

$$s_{ij} = \frac{x_i^\top \cdot x_j}{\|x_i\| \cdot \|x_j\|}, \tag{1}$$

where $x_i, x_j \in \mathbb{R}^d$ represent two data samples in $d$-dimensional column vectors and $\|\cdot\|$ denotes the $\ell^2$-norm of a vector, $s_{ij}$ denotes the entry in the $i$-th row and $j$-th column of the similarity matrix $S$. In the context of similarity search applications, cosine similarity is often used with non-negative vectors [2, 9]. In such cases, the similarity scores range in $(0, 1]$ that can be interpreted as the probability [8] that the pairwise data samples are related. Note, when either $x_i$ or $x_j$ is a zero vector, $s_{ij}$ is not well-defined. To avoid this problem, the widely used method is adding a small value $\epsilon$, i.e., $x_i = \epsilon I$, where $I$ is an identity vector.

*2.1.2 Similarity Matrix Property.* A similarity matrix is symmetric if the similarity measure is symmetric, i.e., Cosine Similarity. Besides, the similarity matrix is PSD if and only if all the eigenvalues generated by singular value decomposition (SVD) [50] are non-negative [28, 29]. Since the similarity matrix is symmetric and PSD, it is also diagonalizable [26]. As a result, its eigenvalues are orthogonal and the SVD of $S$ can be calculated as:

$$S = U\Lambda U^\top = U\Lambda^{\frac{1}{2}}\Lambda^{\frac{1}{2}}U^\top = (U\Lambda^{\frac{1}{2}})(U\Lambda^{\frac{1}{2}})^\top = AA^\top,$$

where $U$ is an orthogonal matrix with columns of $U$ are eigenvectors of $S$, and the entries of a diagonal matrix $\Lambda = diag(\lambda_1, \lambda_2, ..., \lambda_n)$ are non-negative eigenvalues of $S$ based on PSD property. For ease of representation, $A = U\Lambda^{\frac{1}{2}}$, and the corresponding $A^{-1} = (U\Lambda^{\frac{1}{2}})^{-1} = \Lambda^{-\frac{1}{2}}U^{-1} = \Lambda^{-\frac{1}{2}}U^\top = \Lambda^{-1}A^\top$.

### 2.2 Similarity Matrix Approximation

In real scenarios, data corruption is an inevitable problem encountered in most scientific and engineering disciplines. For example, if the query is being sent over a network and there's an unexpected interruption, only a part of the data features in the query item can be transmitted. Besides, if the user is entering a query via a graphical interface, such as the online dialog system,

they might accidentally submit it before finishing typing. To better process the query with incomplete observations and get a more accurate search result, a series of algorithms have been proposed to approximate the incomplete query with missing features, mainly falling into two categories, Missing Value Imputation Methods and Matrix Calibration methods. However, when a dataset contains a small amount of missing data, that is, less than 10% or 15% of the entire dataset, the missing data can be eliminated directly [1, 51]. Otherwise, imputing the data samples with missing observations should be taken into serious consideration.

*2.2.1 Missing Value Imputation (MVI) Methods.* One of the popular methods is the Missing Value Imputation method [14, 39, 40], which is a series of techniques based on statistical principles to replace the missing data with some surrogate value. Mean imputation [32] replaces the missing values with the mean value for the continuous variables, and zero imputation [32] replaces the missing values with 0 without considering the domain knowledge. Besides, pair-wise deleting missing values [24] and list-wise deleting missing values only preserve the values without considering the data distribution causing large information loss and thus leading to a bad performance in real scenarios. In practice, to get a higher-quality estimated dataset, model-based imputation methods, such as linear regression (LR) imputation [49] utilized a linear regression model where the variables with missing data are the dependent variables, and it is predicted using other variables. Meanwhile, the Expectation-Maximization algorithm [43] employed an iterative method to estimate the missing values by maximizing the likelihood estimate of the observed data. Though MVI methods are easy to implement and enjoy high efficiency, accuracy is the bottleneck that limits their applications in practice.

*2.2.2 Matrix Calibration (MC) Methods.* An alternative way to estimate the incomplete similarity matrix is the Matrix Calibration method, which refers to the process of adjusting a matrix-based measurement system to improve its accuracy, which can be done in various scientific and engineering disciplines, especially when dealing with instruments that measure properties using matrix representations. Instead of imputing values into row vectors or column vectors directly, the MC methods forced the similarity matrix to satisfy a specific property. For example, thresholding methods [10, 11] are forced to remove the noise or keep the most important features. Matrix factorization [18, 47] techniques, i.e., singular value decomposition (SVD) [50] can be used to approximate a similarity matrix using fewer informative dimensions. Another series of MC methods [37, 38] focused on studying the specific property, such as symmetric property or positive semi-definiteness (PSD) property [42, 53]. Under this assumption, these methods return an approximation to the target matrix with minimal information loss. However, these methods solve the incomplete problem by processing the entire matrix, which causes a large computational cost and limits the usage of the large-scale dataset. The computational and storage costs motivate us to design a more efficient way to solve the matrix calibration problem.

## 3 PROBLEM FORMULATION

In this section, we formulate the matrix calibration problem based on the insight that both the PSD property and symmetric property can be adapted to solve the matrix calibration problems.

### 3.1 Similarity Matrix Initialization

Given a set of incomplete data samples, the initial similarity matrix $S^0$ is calculated by:

$$s_{ij}^0 = \frac{x_i'^\top \cdot x_j'}{||x_i'|| \cdot ||x_j'||}. \tag{2}$$

Here, $x'$ denotes the incomplete data samples. The approximated cosine similarity value is calculated based on the common features that are observed in both $x_i$ and $x_j$.

### 3.2 Similarity Matrix Calibration Problem

In the similarity matrix calibration problem, we aim to find an estimated $\hat{S}$ has the minimum differences to the initial similarity matrix $S^0$ under a specific constraint:

$$\hat{S} = \arg\min_S ||S - S^0||_F^2$$
$$s.t. \ S \succeq 0, \tag{3}$$

where $S$ is a real symmetric matrix $S_{ii} = 1$ and $0 < S_{ij} = S_{ji} \leq 1$ ($1 \leq i \neq j \leq n$) denoting the similarity score of pairwise data samples, and $|| \cdot ||_F^2$ denotes the squared Frobenius norm of a matrix.

THEOREM 3.1. $||S^* - \hat{S}||_F^2 \leq ||S^* - S^0||_F^2$. The equality holds if and only if $S^0 \in \mathcal{T}$, i.e., $S^0 = \hat{S}$.

Given a dataset $X$ with $n$ samples $\{x_1, \cdots, x_n\}$, and denote $S^* = \{s_{ij}^*\}$ as the ground truth of the similarity matrix, where $s_{ij}^* = s_{ji}^*$ ($1 \leq i, j \leq n$) is the true similarity value between two samples $x_i$ and $x_j$. To simplify the discussion and without loss of generality, we assumed the usage of cosine similarity measure as defined in Eq. (1), while the work applies to a series of other similarity functions or kernels as well [35, 48]. Due to the existence of missing values or observation noises, the true matrix $S^*$ is unknown. As a result, we only have an approximate similarity matrix $S^0$ defined in Eq. (2), that is obtained from incomplete data samples or based on domain-specific knowledge. In this paper, we aim to estimate the initial similarity matrix $S^0$ to approximate the unknown ground truth $S^*$ under the certain PSD property [5, 44, 48].

To simplify the calculation of similarity matrix calibration, we reformulated the target problem in Eq.(3). Given $m$ incomplete query items $Q = \{q_1, ..., q_m\} \in \mathbb{R}^{d \times m}$ in $d$-dimensional space and $n$ complete search candidates $P = \{p_1, ..., p_n\} \in \mathbb{R}^{d \times n}$ in $d$-dimensional space, we aim to estimate the similarity matrix $S \in \mathbb{R}^{(n+m) \times (n+m)}$ to measure the pairwise similarities between query items and search candidates. Here, the initial similarity matrix $S^0$ is calculated by Eq.(2) and can be divided into four sub-matrices:

$$S^0 = \begin{bmatrix} S_{pp} & S_{pq}^0 \\ S_{pq}^{0\top} & S_{qq}^0 \end{bmatrix} \in \mathbb{R}^{(n+m) \times (n+m)},$$

where $S_{pp} \in \mathbb{R}^{n \times n}$ denotes the accurate similarity matrix between $n$ search samples, $S_{pq}^0 \in \mathbb{R}^{n \times m}$ denotes the initial similarity matrix between $n$ search samples and $m$ query samples, $S_{pq}^{0\top} \in \mathbb{R}^{m \times n}$

denotes the transpose of $S_{pq}$, and $S_{qq}^0 \in \mathbb{R}^{m \times m}$ denotes the initial similarity matrix between $m$ query samples. Therefore, the problem in Eq.(3) can be reformulated as:

$$\min_{S_{pq}, S_{qq}} || \begin{bmatrix} S_{pp} & S_{pq} \\ S_{pq}^\top & S_{qq} \end{bmatrix} - \begin{bmatrix} S_{pp} & S_{pq}^0 \\ S_{pq}^{0\top} & S_{qq}^0 \end{bmatrix} ||_F^2$$
$$s.t. \ S \succeq 0, \tag{4}$$

where the $S_{pp}$ is a fixed accurate sub-matrix, both $S_{pq}^0$ and $S_{qq}^0$ are the initial sub-matrices that should be well estimated to build up the entire similarity matrix $\hat{S}$. To reduce the computational cost and preserve the PSD property of $S$, we proposed to calibrate the sub-matrices $\hat{S}_{pq}$ and $\hat{S}_{qq}$ by utilizing the positive definiteness property of the accurate $S_{pp}$ to modify the approximation process.

## 4 METHOD

In this section, we design a novel algorithm based on the well-defined matrix calibration problem shown in Eq.(4). However, the computational complexity highly relies on computing the SVD on the initial similarity matrix. To tackle this problem, we present a conjugate gradient (CG) approximate [46] algorithm that avoids SVD computations.

### 4.1 Basic Similarity Vector Calibration Method

Assume there is 1 incomplete query item $q$, the initial similarity matrix can be divided into $S^0 = \begin{bmatrix} S_{pp} & v^0 \\ v^{0\top} & 1 \end{bmatrix} \in \mathbb{R}^{(n+1) \times (n+1)}$, where the $v^0 \in \mathbb{R}^n$ denotes the similarity vector between the incomplete query and all search candidates and 1 denotes the cosine similarity score between $q$ and itself. Then, the target problem becomes to estimate the similarity vector $\hat{v}$ to preserve the PSD property of $\hat{S}$. Combining the PSD property of $S_{pp}$ and the property of Schur complement [55], we further reformulate the problem as:

$$\min_v || \begin{bmatrix} S_{pp} & v \\ v^\top & 1 \end{bmatrix} - \begin{bmatrix} S_{pp} & v^0 \\ v^{0\top} & 1 \end{bmatrix} ||_F^2$$
$$s.t. \ S_{pp} \succ 0, \ v^\top S_{pp}^{-1} v \leq 1, \tag{5}$$

where the fixed $S_{pp}$ is accurate calculated by $n$ complete search candidate, and $v$ denote the similarity between the given incomplete query item $q$ and the $n$ complete search candidates.

THEOREM 4.1. Let $S_n \in \mathbb{R}^{n \times n}$ be a strictly positive definite matrix. Let $S_{n+1} = \begin{bmatrix} S_n & v \\ v^\top & 1 \end{bmatrix}$, where $v \in \mathbb{R}^n$. Then $S_{n+1}$ is positive semi-definite if and only if $v^\top S_n^{-1} v \leq 1$.

Obviously, the matrix calibration problem can be transferred to the vector calibration problem equivalently by ignoring the $S_{pp}$:

$$\min_{v \in \mathbb{R}^n} ||v - v^0||^2$$
$$s.t \ v^\top S_{pp}^{-1} v \leq 1. \tag{6}$$

To solve this problem, we design a simple yet efficient similarity vector calibration algorithm to recover the similarity vector.

Applying $A = U\Lambda^{\frac{1}{2}}$ and $A^{-1} = \Lambda^{-1}A^\top$ in Section 2.1.2, the objective function and constraint in Eq.(6) can be written as:

$$||v - v^0||^2 = ||AA^{-1}v - AA^{-1}v^0||^2$$
$$= ||A(A^{-1}v - A^{-1}v^0)||^2$$
$$= (A^{-1}v - A^{-1}v^0)^\top A^\top A(A^{-1}v - A^{-1}v^0)$$
$$= (A^{-1}v - A^{-1}v^0)^\top \Lambda (A^{-1}v - A^{-1}v^0),$$
$$v^\top S_{pp}^{-1} v = v^\top (AA^\top)^{-1} v$$
$$= v^\top (A^\top)^{-1} A^{-1} v$$
$$= v^\top (A^{-1})^\top A^{-1} v$$
$$= (A^{-1}v)^\top (A^{-1}v).$$

Now we observe that the variables $v$ and $v^0$ can be changed into $u = A^{-1}v$ and $u^0 = A^{-1}v^0$ to obtain a more concise form of convex optimization problem:

$$\min_{v \in \mathbb{R}^n} (A^{-1}v - A^{-1}v^0)^\top \Lambda (A^{-1}v - A^{-1}v^0)$$
$$s.t. \ (A^{-1}v)^\top (A^{-1}v) \leq 1.$$
$$\leftrightarrow \min_{u \in \mathbb{R}^n} \frac{1}{2}(u - u^0)^\top \Lambda (u - u^0) \tag{7}$$
$$s.t. \ u^\top u \leq 1.$$

To solve this optimization problem, we consider two cases:
1) If $u^{0\top} u^0 \leq 1$, then $u^* = u^0$ is the solution. The $\hat{v} = Au^0$
2) If $u^{0\top} u^0 > 1$, then the solution appears on the boundary.
For the second case, we define the Lagrangian function as:

$$L \equiv \frac{1}{2} u^\top \Lambda u - u^\top \Lambda u^0 + \gamma(u^\top u - 1) \text{ for } \gamma \geq 0 \tag{8}$$

From the KKT condition [23] of convex optimization, we have:

$$\begin{cases} \frac{\partial L}{\partial u} = \Lambda u - \Lambda u^0 + 2\gamma u = 0 \\ \gamma(u^\top u - 1) = 0 \\ \gamma \geq 0 \\ u^\top u - 1 \leq 0 \end{cases} \Rightarrow \begin{cases} u = (\Lambda + 2\gamma I)^{-1} \Lambda u^0 \\ u^\top u = 1. \end{cases} \tag{9}$$

Fortunately, the estimated $u$ can be denoted by the function of $\gamma$, which means $u$ and $\gamma$ can be calculated iteratively.

Specifically, the $\hat{u}$ is denoted by $\gamma$ and can be adjusted by varying the value of $\gamma$, that is, when the constraint is not satisfied, the algorithm prefers to choose the smaller $\gamma$ to meet the constraint and further calculate the $\hat{u}$. Finally, the $\hat{v}$ is calculated by: $\hat{v} = A\hat{u}$.

We summarized the basic idea of Basic Similarity Vector Calibration (BSVC) Methods in Algorithm.1. Line 1 shows the SVD on the accurate similarity matrix $S_{pp}$. Line 2 shows the calculation of the auxiliary variable $A$ for ease of calculation. Line 3 shows the initialization of similarity vector $v^0$ and auxiliary vector $u^0$. Lines 4-18 show the calculation of optimal $\hat{u}$ and the corresponding $\gamma$. Line 19 returns the estimated similarity vector $\hat{v}$.

Moreover, the searching time of $\gamma$ is dependent on the range $(\lambda_{max}, \lambda_{min})$. To reduce the computation cost, we adopt a Quasi-Newton (QN) [16] method to calculate the $\gamma$. Based on the constraint, $u$ can be written as a function of $\gamma$:

$$f(\gamma) = ||((\Lambda + 2\gamma I)^{-1} \Lambda u^0)^\top (\Lambda + 2\gamma I)^{-1} \Lambda u^0)) - 1||^2. \tag{10}$$

---

**Algorithm 1 BSVC: Basic Similarity Vector Calibration Method**

---

**Input:** $S_{pp} \in \mathbb{R}^{n \times n}$: Accurate similarity matrix of $n$ complete search samples; $v^0 \in \mathbb{R}^n$: Initial similarity vector between 1 incomplete query item and $n$ complete search samples; $\gamma$: Lagrange multiplier; $tol = 10^{-4}$: Tolerance.

**Output:** $\hat{v} \in \mathbb{R}^n$: Estimated similarity vector with a unique $\hat{\gamma}$.

1: SVD on $S_{pp} = U\Lambda U^\top$, where eigenvalues are sorted in a descending order $\lambda_{max}, ... \lambda_{min}$;
2: Calculate $A = U\Lambda^{\frac{1}{2}}$ and $A^{-1} = \Lambda^{-1}A^\top$;
3: Initialize $v^0$ by Eq.(2) and correlated $u^0 = A^{-1}v^0$;
4: **if** $\hat{u}^\top \hat{u} <= 1 + tol$ **then**
5:      $\hat{v} = Au^0$;
6: **else**
7:      Calculate $\gamma_{min}$ s.t. $||u||_2^2 = \sum_{i=1}^n (\frac{\lambda_i u_i^0}{\lambda_{max} + 2\gamma_{min}})^2 = 1$;
8:      Calculate $\gamma_{max}$ s.t. $||u||_2^2 = \sum_{i=1}^n (\frac{\lambda_i u_i^0}{\lambda_{min} + 2\gamma_{max}})^2 = 1$;
9:      **while** $\hat{u}^\top \hat{u} > 1 + tol$ **do**
10:          Update $\hat{\gamma} = 0.5 * (\gamma_{min} + \gamma_{max})$;
11:          Update $\hat{u} \leftarrow (\Lambda + 2\hat{\gamma}I)^{-1} \Lambda u^0$;
12:          **if** $\hat{u}^\top \hat{u} > 1$ **then**
13:              Update $\gamma_{min} \leftarrow \hat{\gamma}$;
14:          **else**
15:              Update $\gamma_{max} \leftarrow \hat{\gamma}$;
16:          **end if**
17:      **end while**
18: **end if**
19: Calculate $\hat{v} = A\hat{u}$;
20: **return** $\hat{v}$

---

More detailed, QN provided a way to approximate the optimal solution $\gamma$ to achieve the optimal value of $f(\gamma)$, which builds up an approximation to the inverse Hessian matrix to achieve the optimal solution. The algorithm is stopped when the gradient is smaller than a certain tolerance, or a maximum number of iterations is reached, stop the algorithm and return the approximated $\hat{\gamma}$ as the result.

## 4.2 Approximated Basic Similarity Vector Calibration Method

Though the BSVC method provides a set of solutions of $\gamma$ and $v$, the computational complexity of SVD on the initial similarity matrix $S_{pp}$ as $O(n^3)$ is the bottleneck when the size of $S_{pp}$ is extremely large. To reduce the computational cost, we propose to approximate the SVD process and rewrite the problem in Eq.(6) as:

$$\min_{v \in \mathbb{R}^n} \frac{1}{2}(v - v^0)^\top (v - v^0)$$
$$s.t \ v^\top S_{pp}^{-1} v \leq 1. \tag{11}$$

Similarly, we consider two cases:
1) If $v^{0\top} S_{pp}^{-1} v^0 \leq 1$, then $\hat{v} = v^0$ is the solution.
2) If $v^{0\top} S_{pp}^{-1} v^0 > 1$, then the solution appears on the boundary.
Also, for the second case, we define the Lagrangian function as:

$$\tilde{L} \equiv \frac{1}{2} v^\top I v - v^\top I v^0 + \tilde{\gamma}(v^\top S_{pp}^{-1} v - 1) \text{ for } \tilde{\gamma} \geq 0.$$

Similar to the solution in Eq.(9), we have:

$$
\begin{cases}
\dfrac{\partial \tilde{L}}{\partial v} = Iv - Iv^0 + 2\tilde{\gamma}S_{pp}^{-1}v = 0 \\
\tilde{\gamma}(v^\top S_{pp}^{-1}v - 1) = 0 \\
\tilde{\gamma} \geq 0 \\
v^\top S_{pp}^{-1}v - 1 \leq 0
\end{cases}
\Rightarrow
\begin{cases}
v = (I + 2\tilde{\gamma}S_{pp}^{-1})^{-1}v^0 \\
v^\top S_{pp}^{-1}v = 1.
\end{cases}
\tag{12}
$$

Rather than directly calculate the $S_{pp}^{-1}$, which might be computationally expensive and unstable with the extremely large size of the similarity matrix $S_{pp}^{-1}$. We leverage the Conjugate Gradient (CG) [46] algorithm to find an approximated $S_{pp}^{-1}$, that is, solving the problem $S_{pp}X = I$ by CG. Here, $I$ is the identity matrix, where the diagonal elements are 1. However, the conventional CG method cannot handle the matrix calculation, we can solve the linear problem $S_{pp}x = e$, where $S_{pp}$ is a symmetric positive definite matrix, $e$ represents the $i$-th column of identity matrix $I$, and the $i$-th element is 1. Then, the $S_{pp}^{-1}$ is generated by vector calculation. Consequently, the calculation of $v$ can be simplified by Taylor expression [27] to avoid the inverse calculation.

$$
(I + 2\tilde{\gamma}S_{pp}^{-1})^{-1} = I^{-1} - I^{-1}2\tilde{\gamma}S_{pp}^{-1}I^{-1} + I^{-1}2\tilde{\gamma}S_{pp}^{-1}I^{-1}2\tilde{\gamma}S_{pp}^{-1}I^{-1} - \dots
$$
$$
\approx I - 2\hat{\gamma}S_{pp}^{-1}.
$$

Here, we retain the first two terms, the approximated update equation is:

$$
v \approx (I - 2\tilde{\gamma}S_{pp}^{-1})v^0
\tag{13}
$$

Then, the $\hat{v}$ and the correlated $\hat{\tilde{\gamma}}$ can be updated iteratively. Similarly, $\tilde{\gamma}$ can be solved by QN and the objective function as:

$$
f(\tilde{\gamma}) = \|((I - 2\tilde{\gamma}S_{pp}^{-1})v^0)^\top S_{pp}^{-1}((I - 2\tilde{\gamma}S_{pp}^{-1})v^0)) - 1\|^2
\tag{14}
$$

Algorithm 2 summarized the basic idea of the CQABSVC. Line 1 shows the initialization of $v^0$ and $\tilde{\gamma}$. Lines 2-5 show the calculation of $S_{pp}^{-1}$ by CG algorithm. Lines 6-13 show the calculation of $\hat{v}$ and $\tilde{\gamma}$. Line 14 returns the $\hat{v}$.

### 4.3 Similarity Matrix Calibration Method

Due to the independence between any $v_i$ and $v_j$, the similarity matrix $\hat{S}_{pq} = \{v_1, ..., v_m\} \in \mathbb{R}^{n \times m}$ can be calibrated $m$ times by BSVC when the number of query samples $m > 1$. Following the BSVC method, these $m$ independent vectors $\{v_i\}_{i=1}^m$ can be calibrated by BSVC/ABSVC via Algorithm.1 /Algorithm.2 sequentially. We summarize the similarity matrix calibration method in Algorithm.3, which is the main algorithm in this paper. Line 1 shows the initialization of similarity $S^0$. Lines 2-5 show the calculation via Algorithm.1/2. Line 6 returns the estimated $\hat{S}$.

## 5 ANALYSIS

### 5.1 Memory Requirement

Given a set of search database $P \in \mathbb{R}^{d \times n}$ with $n$ complete data samples in $d$ dimensional space and a query database $Q \in \mathbb{R}^{d \times m}$ with $m$ incomplete data samples in $d$ dimensional space, we need $O(dn)$ and $O(dm)$ to store the search database and the query database. Meanwhile, we need $O((n + m)^2)$ to store the similarity

---

**Algorithm 2 CQABSVC: C**onjugate Gradient and **Q**usai-Newton based **A**pproximate **B**asic **S**imilarity **V**ector **C**alibration Method

**Input:** $S_{pp} \in \mathbb{R}^{n \times n}$: Similarity matrix of $n$ complete search samples; $v^0 \in \mathbb{R}^n$: Initial similarity vector between 1 incomplete query item and all complete search samples; $\tilde{\gamma}$: Lagrange multiplier; $tol$: Tolerance for CQABSVC; $tol_1 = tol_2 = 10^{-4}$: Tolerance for CG and QN; $I \in \mathbb{R}^{n \times n}$: Identity Matrix; $e$: $i$-th column of $I$;

**Output:** $\hat{v} \in \mathbb{R}^n$: Estimated similarity vector.
1:  Initialize $v^0$ by Eq.(2) and $\tilde{\gamma} = 1$;
2:  **for** $i = 1 : n$ **do**
3:      Solve $S_{pp}X(:, i) = e$ via CG to get the $i$-th column of $S_{pp}^{-1}$;
4:  **end for**
5:  Get $X = S_{pp}^{-1}$;
6:  **if** $\hat{v}^\top S_{pp}^{-1}\hat{v} <= 1 + tol$ **then**
7:      $\hat{v} = v^0$
8:  **else**
9:      **while** $\hat{v}^\top S_{pp}^{-1}\hat{v} > 1 + tol$ **do**
10:          Update $\hat{\gamma}$ by Quasi-Newton method;
11:          Update $\hat{v} \leftarrow (I - 2\hat{\gamma}S_{pp}^{-1})v^0$;
12:      **end while**
13:  **end if**
14:  **return** $\hat{v}$

---

**Algorithm 3** Similarity Matrix Calibration Method (**Main Algorithm**)

**Input:** $S_{pp} \in \mathbb{R}^{n \times n}$: Similarity matrix of $n$ complete search samples; $S_{pq}^0 \in \mathbb{R}^{n \times m}$: Approximate similarity matrix between $n$ complete search samples and $m$ incomplete query samples;

**Output:** $\hat{S} \in \mathbb{R}^{(n+m) \times (n+m)}$: Estimated similarity matrix of $n$ complete search samples and $m$ incomplete query samples.
1:  Initialize $S_{pq}^0$ by Eq.(2);
2:  **for** $i = 1, ..., m$ **do**
3:      $\hat{v} \leftarrow$ calibrate $v^0$ by Algorithm.1/2;
4:      Update $i^{th}$ similarity vector $\hat{v}$;
5:  **end for**
6:  **return** $\hat{S}$

---

matrix $S \in \mathbb{R}^{(n+m) \times (n+m)}$. In total, the memory requirement is $O(d(n + m) + (n + m)^2)$.

### 5.2 Computational Complexity Analysis

For the Matrix calibration methods, DMC applied SVD on the entire matrix with computational complexity $O((n + m)^3)$ in each iteration. CMC firstly divided the entire matrix into $r$ sub-matrices and calibrated the sub-matrices by an iterative projection method. Our SMC required $O((n + m)^3)$ to decompose the initial similarity matrix $S_{pp}$, $O((n + m)^2)$ to calculate the $u$, $O(\log_2(n + m))$ to find the unique $\hat{\gamma}$ via bisection algorithm. For the QNASMC, we need $O(n^3)$ to perform SVD, $O((n + m)^2)$ to calculate the $v$, $O(kmn)$ to find the unique $\hat{\gamma}$ via QN algorithm. For QNASMC, we need $O(n^3)$ to perform inverse operation on $S_{pp}$, $O((n + m)^2)$ to calculate the $v$, $O(kmn)$ to find the unique $\hat{\gamma}$ via QN algorithm. The total complexity

is $O(n^2(k_1 + k_2))$, which is much smaller than the other baseline methods.

## 5.3 Convergence

THEOREM 5.1. *When running conjugate gradient with $k_1$ step, and Quasi-Netwon method with $k_2$ step, CQABSVC has*

$$||v^* - \left(I + 2\tilde{\gamma}' S_{pp}^{-1}{}'\right)^{-1} v^0|| \leq O(c_0^{k_1}) + O(c_1^{k_2})$$

*where $0 < c_0 < 1$, and $0 < c_1 < 1$.*

$v^* = \left(I + 2\gamma^* S_{pp}^{-1}\right)^{-1} v^0$ by the KKT condition, where $\gamma^*$ satisfying $(\left(I + 2\tilde{\gamma} S_{pp}^{-1}\right)^{-1} v^0)^\top S_{pp}^{-1} \left(I + 2\tilde{\gamma} S_{pp}^{-1}\right)^{-1} v^0 = I$.

## 6 EXPERIMENTS

We conducted exhaustive matrix calibration experiments on both visual datasets and textual datasets. We applied effectiveness, efficiency, and sensitivity to evaluate the capability of similarity matrix calibrations. All experiments are repeated 5 runs, and the average performance is reported.

## 6.1 Experimental Settings

*6.1.1 Datasets.* We adopted the following datasets to verify the generality of the main algorithm SMC: **ImageNet** [15]: an image dataset across more than $20,000$ categories with $1,000$ diemensions. **MNIST** [34]: a grayscale image dataset of handwritten digits (e.g. 0-9) with 784 dimensions. **CIFAR10** [33]: an image dataset of 10 real objects with $1,024$ dimensions. **SIFT** [20]: an image dataset of SIFT features with 128 dimensions. **RCV1** [36]: a textual dataset of newswire stories from Reuters with $47,236$ dimensions.

*6.1.2 Baselines.* We adopted the following baseline methods, including Missing Value Imputation (MVI) methods and Matrix Calibration (MC) methods. For all methods that we compare in the experiments, we use public codes unless they are not available.

- **Missing Value Imputation Methods**: **Mean Imputation (MEAN), ZERO,** $k$**NN** [32]: The missing values are imputed by the mean value, 0, or $k$-nearest neighbor values in search data samples, respectively. **Linear Regression (LR)** [49]: The missing values are imputed by the multivariate linear regression between observed features and missing features. **GROUSE** [7]: GROUSE imputed the missing value based on low-rank matrix completion. **KFMC** [19]: KFMC imputed and optimized the missing value in online data by high-rank matrix completion.
- **Matrix Calibration Methods**: **Direct Matrix Calibration (DMC)** [37]. DMC calibrated the similarity matrix by searching the approximated matrix with PSD based on Dykstra's alternating projection method. **Cyclic Matrix Calibration (CMC)** [38]. CMC proposed a cyclic projection method to seek a similarity matrix satisfied with PSD property.

All the baselines are implemented in MATLAB. Each algorithm is stopped when the relative difference between objective values in consecutive iterations is smaller than the tolerance $10^{-4}$. We evaluated the performance varying with the missing ratios by grid

search and adopted the hyperparameter as mentioned in the respective papers. We denoted our main method in Algorithm.3 as SMC, QNASMC, and CQASMC, where the similarity vector was estimated by BSVC, QNBSVC, and CQABSVC.

## 6.2 Effectiveness Analysis

We applied relative-mean-square error (RMSE) [3] as the evaluation metric to verify the effectiveness of the comparison method, which is defined as:

$$\text{RMSE} = \frac{||\hat{S} - S^*||_F^2}{||S^0 - S^*||_F^2}$$

Table 1 showed the average RMSE and STD (standard deviation) varied with various missing ratios $\rho = \{20\%, 50\%, 80\%\}$ on different datasets. Overall, all the MC methods had a performance guarantee with RSME $\leq 1$ while the MVI methods could not achieve this, which revealed the necessity of utilizing the prior knowledge of the similarity matrix. Take ImageNet as an example, the RMSE of our SMC was 0.401 with a missing ratio $\rho = 0.8$, which was much better than all the other baseline methods. The same tendencies could also be found on other datasets. Mathematically, the larger $\rho$ denoted the initial approximated $S^0$ calculated by the incomplete data samples is far away from the ground truth $S^*$ calculated by complete data samples, leaving a higher probability of improving matrix calibration results. Combining the performance on all the datasets with various missing ratios $\rho$, the RMSE of our proposed methods changed steadily. Therefore, a more evident calibration results of $||\hat{S} - S^*||_F^2$ was achieved through calibration for a higher missing ratio $\rho$. More results of RMSE are in Appendix.A.

## 6.3 Efficiency Analysis

Table 2 recorded the time of all the baseline methods with $m = 1,000$ query items and $n = 1,000/n = 5,000$ search candidates. Since all the MVI methods ignored the pairwise similarities of the datasets, though the time cost was much lower than MC methods, the RMSE were worse than MC methods. Here, we ignored the comparison between MVI methods and MC methods. Take ImageNet as an example, DMC, CMC, and SMC took around 100 seconds, which was possibly caused by the SVD operation. Fortunately, the time cost of QASMC and CQASMC were reduced by using the approximated algorithm, to around 70 seconds and 30 seconds, respectively. Meanwhile, the same tendency can be found on the other datasets. More results of Time are in Appendix.A.

## 6.4 Evaluation for PSD property

To verify the assumption, that is, the positive definiteness of similarity matrix $S_{pp}$ always holds, we also analyzed the eigenvalues of the initial accurate similarity matrix $S_{pp}$ that was calculated by the search samples. For ease of representation, we showed the maximum and minimum eigenvalues of $S_{pp}$. As can be seen from Table 3, the maximum eigenvalue varied with various data sizes. It seemed that the similarity matrix $S_{pp}$ with a larger size gained a larger eigenvalue. Meanwhile, all the minimum eigenvalues were positive, which can be seen as a guarantee that the similarity matrix $S_{pp}$ was positive definite.

**Table 1: Comparison of RMSE with $m = 1,000$ query items and $n = 1,000$ query items with various missing $\rho$ on different datasets.**

| $n{=}1{,}000$ $m{=}1{,}000$ | Missing Ratio $\rho$ | ZERO | MEAN | kNN | LR | RF | GROUSE | KFMC | DMC | CMC | SMC | QASMC | CQASMC |
|---|---|---|---|---|---|---|---|---|---|---|---|---|---|
| ImageNet | 0.2 | 2027.300±371.24 | 1.600±0.045 | 1.306±0.035 | 134.580±14.586 | 1.219±0.051 | 1.535±0.195 | 0.951±0.057 | 0.735±0.021 | 0.799±0.004 | 0.706±0.087 | **0.691**±0.004 | **0.630**±0.114 |
|  | 0.5 | 3421.600±499.21 | 2.100±0.357 | 1.639±0.357 | 80.897±24.180 | 1.716±0.016 | 1.975±0.371 | 1.131±0.091 | 0.618±0.012 | 0.620±0.044 | 0.586±0.018 | **0.472**±0.014 | **0.417**±0.076 |
|  | 0.8 | 2785.200±390.21 | 1.171±0.127 | 0.875±0.020 | 66.013±15.411 | 1.046±0.067 | 0.999±0.025 | 0.549±0.063 | 0.451±0.042 | 0.427±0.032 | **0.401**±0.010 | 0.398±0.043 | 0.432±0.077 |
| MNIST | 0.2 | 1972.000±193.601 | 1.747±0.018 | 1.470±0.062 | 128.260±19.243 | 1.354±0.002 | 1.675±0.078 | 1.093±0.058 | 0.742±0.017 | 0.804±0.008 | **0.738**±0.060 | 0.731±0.02 | 0.745±0.063 |
|  | 0.5 | 3691.100±125.221 | 2.159±0.069 | 1.742±0.009 | 88.051±31.935 | 1.728±0.028 | 2.008±0.002 | 1.131±0.092 | 0.581±0.073 | 0.587±0.028 | 0.549±0.024 | **0.520**±0.01 | 0.532±0.068 |
|  | 0.8 | 2748.009±233.681 | 1.220±0.039 | 0.936±0.058 | 4.596±13.007 | 1.093±0.037 | 1.048±0.024 | 0.579±0.045 | 0.441±0.048 | **0.416**±0.005 | 0.421±0.010 | 0.465±0.004 | 0.486±0.014 |
| CIFAR | 0.2 | 3180.278±193.410 | 2.639±0.073 | 1.951±0.009 | 79.778±15.991 | 2.027±0.033 | 2.434±0.075 | 1.346±0.095 | 0.772±0.017 | 0.787±0.013 | **0.743**±0.029 | 0.763±0.03 | **0.715**±0.052 |
|  | 0.5 | 3382.103±214.877 | 1.828±0.063 | 1.408±0.056 | 69.523±14.488 | 1.460±0.086 | 1.720±0.079 | 1.014±0.054 | **0.590**±0.043 | 0.591±0.069 | **0.590**±0.064 | 0.595±0.09 | **0.465**±0.047 |
|  | 0.8 | 2883.282±154.379 | 1.110±0.375 | 0.840±0.079 | 4.420±19.439 | 1.002±0.063 | 0.963±0.039 | 1.560±0.015 | **0.378**±0.047 | 0.380±0.036 | **0.379**±0.004 | 0.390±0.04 | 0.459±0.044 |
| SIFT | 0.2 | 1797.553±100.341 | 1.068±0.092 | 0.724±0.017 | 4.517±14.233 | 0.909±0.022 | 0.872±0.064 | 1.495±0.004 | 0.566±0.057 | **0.557**±0.008 | 0.575±0.044 | 0.598±0.025 | **0.474**±0.069 |
|  | 0.5 | 2425.479±212.097 | 0.957±0.016 | 0.724±0.081 | 5.971±16.069 | 0.835±0.021 | 0.806±0.026 | 1.489±0.093 | 0.486±0.051 | 0.460±0.096 | **0.432**±0.080 | 0.447±0.022 | 0.547±0.086 |
|  | 0.8 | 3382.103±154.482 | 1.828±0.046 | 1.408±0.083 | 69.523±15.411 | 1.460±0.025 | 1.720±0.027 | 1.014±0.096 | 0.607±0.085 | 0.605±0.059 | **0.576**±0.025 | 0.589±0.023 | **0.431**±0.077 |
| PROTEIN | 0.2 | 2789.414±167.955 | 1.051±0.089 | 1.034±0.062 | 64.234±10.520 | 1.909±0.092 | 2.412±0.088 | 1.541±0.089 | **0.514**±0.007 | 0.624±0.008 | **0.590**±0.078 | 0.613±0.029 | 0.715±0.065 |
|  | 0.5 | 2579.437±191.385 | 1.075±0.034 | 0.848±0.086 | 63.414±31.581 | 0.966±0.056 | 0.922±0.025 | 1.493±0.046 | **0.445**±0.081 | 0.601±0.083 | 0.557±0.099 | 0.568±0.024 | **0.404**±0.040 |
|  | 0.8 | 3382.103±90.545 | 1.828±0.019 | 1.408±0.083 | 69.523±37.103 | 1.460±0.008 | 1.720±0.086 | 1.014±0.006 | 0.607±0.069 | 0.605±0.047 | **0.572**±0.097 | 0.576±0.024 | **0.460**±0.065 |
| RCV1 | 0.2 | 2561.412±97.155 | 1.042±0.171 | 1.241±0.070 | 64.141±10.497 | 1.491±0.006 | 2.131±0.005 | 1.416±0.078 | 0.624±0.061 | 0.612±0.073 | **0.583**±0.037 | 0.592±0.002 | 0.699±0.042 |
|  | 0.5 | 2835.411±129.567 | 1.218±0.902 | 0.921±0.090 | 101.476±12.578 | 1.940±0.022 | 2.351±0.008 | 1.153±0.071 | 0.527±0.018 | 0.540±0.051 | **0.480**±0.042 | 0.499±0.023 | **0.456**±0.004 |
|  | 0.8 | 2967.128±126.516 | 1.959±0.133 | 1.615±0.044 | 73.165±14.868 | 1.578±0.072 | 1.811±0.075 | 1.134±0.009 | 0.659±0.076 | 0.661±0.088 | **0.615**±0.047 | 0.626±0.004 | **0.527**±0.086 |

**Table 2: Comparison of Time Cost (seconds) with $n = 1,000/n = 5,000$ search candidates and $m = 1,000$ query items with various missing $\rho$ on different datasets. A smaller value denoted better performance.**

| $n{=}1{,}000$ $m{=}1{,}000$ | Missing Ratio $\rho$ | DMC | CMC | SMC | QASMC | CQASMC |
|---|---|---|---|---|---|---|
| ImageNet | 0.2 | 104.243 | 229.271 | 199.110 | 75.605 | 26.593 |
|  | 0.5 | 102.495 | 246.419 | 158.400 | 76.494 | 26.662 |
|  | 0.8 | 105.540 | 305.849 | 139.920 | 77.680 | 27.414 |
| MNIST | 0.2 | 102.018 | 213.748 | 128.548 | 74.756 | 20.042 |
|  | 0.5 | 96.366 | 227.495 | 128.651 | 78.369 | 22.054 |
|  | 0.8 | 100.333 | 223.190 | 133.114 | 79.714 | 24.442 |
| CIFAR | 0.2 | 77.827 | 182.271 | 204.948 | 88.473 | 27.806 |
|  | 0.5 | 77.421 | 230.872 | 136.358 | 83.254 | 26.076 |
|  | 0.8 | 78.035 | 222.290 | 133.547 | 81.146 | 25.989 |
| SIFT | 0.2 | 68.345 | 115.203 | 158.424 | 99.052 | 26.140 |
|  | 0.5 | 74.155 | 144.967 | 294.152 | 94.004 | 27.734 |
|  | 0.8 | 73.617 | 146.608 | 300.467 | 77.110 | 28.342 |
| PROTEIN | 0.2 | 40.279 | 114.849 | 175.261 | 74.846 | 27.846 |
|  | 0.5 | 37.761 | 107.814 | 200.859 | 79.932 | 26.200 |
|  | 0.8 | 44.116 | 122.680 | 156.713 | 82.293 | 26.280 |
| RCV1 | 0.2 | 49.812 | 144.485 | 157.727 | 81.344 | 26.330 |
|  | 0.5 | 46.764 | 107.778 | 168.687 | 69.088 | 27.963 |
|  | 0.8 | 47.326 | 116.575 | 156.085 | 77.404 | 26.459 |

| $n{=}5{,}000$ $m{=}1{,}000$ | Missing Ratio $\rho$ | DMC | CMC | SMC | QASMC | CQASMC |
|---|---|---|---|---|---|---|
| ImageNet | 0.2 | 94.328 | 210.688 | 199.870 | 71.849 | 26.853 |
|  | 0.5 | 96.158 | 271.144 | 137.401 | 78.580 | 26.456 |
|  | 0.8 | 95.490 | 220.163 | 165.499 | 72.797 | 27.370 |
| MNIST | 0.2 | 94.858 | 223.799 | 161.762 | 80.415 | 27.592 |
|  | 0.5 | 94.265 | 220.787 | 165.966 | 77.925 | 26.394 |
|  | 0.8 | 95.594 | 228.628 | 155.679 | 73.224 | 26.131 |
| CIFAR | 0.2 | 91.382 | 227.721 | 197.559 | 64.817 | 27.358 |
|  | 0.5 | 92.469 | 248.798 | 273.604 | 79.933 | 26.138 |
|  | 0.8 | 95.732 | 238.644 | 311.500 | 85.818 | 28.875 |
| SIFT | 0.2 | 76.234 | 111.764 | 201.908 | 81.888 | 27.605 |
|  | 0.5 | 75.511 | 204.754 | 254.438 | 82.472 | 27.517 |
|  | 0.8 | 48.926 | 112.838 | 171.038 | 91.427 | 28.149 |
| PROTEIN | 0.2 | 46.715 | 109.288 | 163.453 | 64.817 | 28.871 |
|  | 0.5 | 43.736 | 104.187 | 194.820 | 79.933 | 27.216 |
|  | 0.8 | 45.799 | 106.984 | 165.449 | 85.818 | 26.508 |
| RCV1 | 0.2 | 47.084 | 112.832 | 155.052 | 89.754 | 28.282 |
|  | 0.5 | 47.333 | 110.628 | 163.994 | 93.327 | 26.318 |
|  | 0.8 | 47.016 | 121.145 | 158.010 | 89.542 | 26.364 |

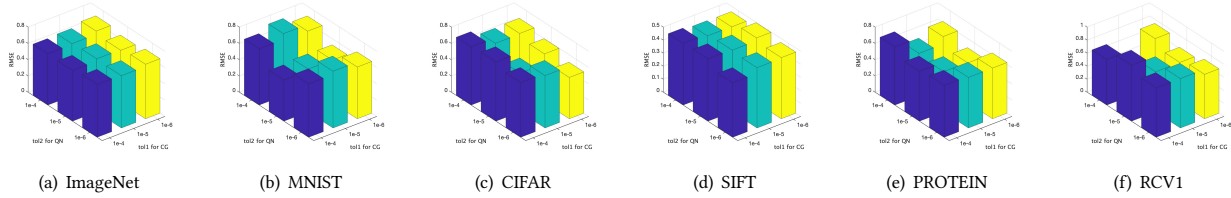

(a) ImageNet    (b) MNIST    (c) CIFAR    (d) SIFT    (e) PROTEIN    (f) RCV1

**Figure 1: Comparisons of RMSE with various settings of tolerance for CG and QN on different datasets. $\rho=0.2$, $m = 1,000$, and $n = 1,000$.**

## 6.5 Evaluation for tolerance of CG and QN

To verify the overall performance of our ASMC, we conducted the ablation study to gain a better understanding of different settings of tolerance of CG and QN, shown in Fig.1-Fig.3. Overall, the RMSE decreased with the increase of missing ratio $\rho$. Meanwhile, RMSE was not much changed with a fixed missing ratio $\rho$ on a specific in most cases. More detailed, when the missing ratio $\rho = 0.2$, the RMSE was around 0.6 on the ImageNet dataset with various settings of tolerance. Similar tendencies can be found on the other datasets, which was probably because the lower missing ratio has a smaller influence on the RMSE. When $\rho = 0.5$, the RMSE on PROTEIN datasets was only 0.145. It was probably caused by the data distribution.

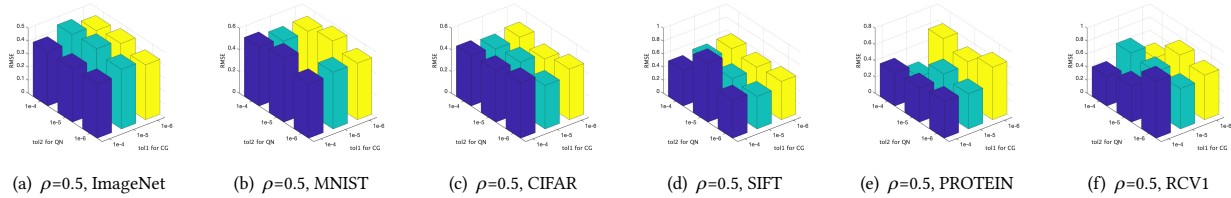

(a) $\rho$=0.5, ImageNet    (b) $\rho$=0.5, MNIST    (c) $\rho$=0.5, CIFAR    (d) $\rho$=0.5, SIFT    (e) $\rho$=0.5, PROTEIN    (f) $\rho$=0.5, RCV1

**Figure 2: Comparisons of RMSE with various settings of tolerance for CG and QN on different datasets, $\rho$=0.5, $m = 1,000$, and $n = 1,000$.**

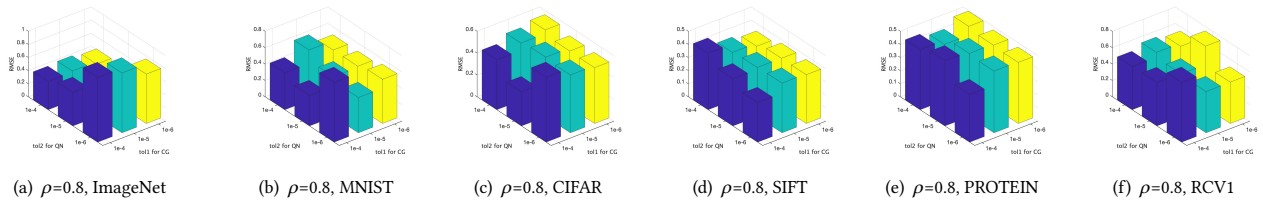

(a) $\rho$=0.8, ImageNet    (b) $\rho$=0.8, MNIST    (c) $\rho$=0.8, CIFAR    (d) $\rho$=0.8, SIFT    (e) $\rho$=0.8, PROTEIN    (f) $\rho$=0.8, RCV1

**Figure 3: Comparisons of RMSE with various settings of tolerance for CG and QN on different datasets, $\rho = 0.8$, $m = 1,000$ and $n = 1,000$.**

**Table 3: Maximum Eigenvalue $\lambda_{\max}$ and Minimum Eigenvalue $\lambda_{\min}$ of Various size of $S_{\mathbf{pp}} \in \mathbb{R}^{n \times n}$.**

| Dataset | Eigenvalue | $n = 1,000$ | $n = 2,000$ | $n = 3,000$ | $n = 4,000$ | $n = 5,000$ |
|---|---|---|---|---|---|---|
| ImageNet | $\lambda_{\max}$ | 756.5818 | 1.51e+03 | 2.27e+03 | 3.03e+03 | 3.79e+03 |
| | $\lambda_{\min}$ | 7.55e-14 | 1.40e-13 | 1.90e-13 | 1.91e-13 | 3.79e-13 |
| MNIST | $\lambda_{\max}$ | 632.1339 | 2.09e+03 | 1.38e+03 | 1.85e+03 | 3.54e+03 |
| | $\lambda_{\min}$ | 2.83e-14 | 2.40e-14 | 4.53e-14 | 9.81e-14 | 5.55e-14 |
| CIFAR | $\lambda_{\max}$ | 759.6139 | 1.52e+03 | 2.28e+03 | 3.04e+03 | 3.80e+03 |
| | $\lambda_{\min}$ | 3.35e-14 | 1.10e-13 | 1.59e-13 | 1.13e-13 | 1.22e-13 |
| SIFT | $\lambda_{\max}$ | 757.9125 | 1.52e+03 | 2.28e+03 | 3.04e+03 | 3.79e+03 |
| | $\lambda_{\min}$ | 7.57e-14 | 1.52e-13 | 2.28e-13 | 1.95e-13 | 2.19e-13 |
| PROTEIN | $\lambda_{\max}$ | 763.8486 | 1.52e+03 | 2.28e+03 | 3.04e+03 | 3.79e+03 |
| | $\lambda_{\min}$ | 7.60e-14 | 1.01e-14 | 1.83e-13 | 2.72e-13 | 2.83e-13 |
| RCV1 | $\lambda_{\max}$ | 760.9445 | 1.52e+03 | 2.27e+03 | 3.03e+03 | 3.79e+03 |
| | $\lambda_{\min}$ | 7.59e-14 | 1.17e-13 | 1.58e-13 | 2.75e-13 | 3.53e-13 |

## 7 CONCLUSION

Matrix calibration is a fundamental research problem with various applications. However, it is often non-trivial to obtain a good similarity matrix when the query samples are incomplete in real-world scenarios. In this paper, we proposed a Basic Similarity Vector Calibration (BSVC) method to solve the similarity search problem with incomplete queries. To further reduce the computational complexity, we proposed an approximated algorithm, Conjugate Gradient and Quasi-Newton based Approximated BSVC(CQABSVC) method to find the approximated solutions. Then the similarity matrix can be calibrated by BSVC or CQASBVC methods sequentially. Theoretical analysis showed the convergence guarantee of our proposed method. Empirical analysis showed a clear improvement in estimating the similarity matrix with superior performance over baseline methods, which provided a promising practical tool in real-world scenarios.

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

## A  EXTENDED ALGORITHM AND EVALUATION

### A.1  RMSE Results

In this section, we showed the comparison of RMSE with $n = 2,000, 4,000, 5,000$ search candidates and $m = 1,000$ query items with various missing $\rho$ on different datasets.

### A.2  Time Results

In this section, we showed the comparison of Time with $n = 2,000, 4,000$ search candidates and $m = 1,000$ query items with various missing $\rho$ on different datasets.

## B  PROOFS

### B.1  Explanation of ASVC

This iteration method provided a way to approximate the roots of a real-valued function. The detailed update in each iteration is:

- Evaluate the gradient $g(\tilde{\gamma})$
- Determine the direction $p_t = -H_t g_t$
- Use a line search to find $\alpha_t$ that minimizes $f(\tilde{\gamma}_t + \alpha_t p_t)$
- Update $\tilde{\gamma}_{t+1} = \tilde{\gamma}_t + \alpha_t p_t$
- Compute $\tilde{\gamma}_{t+1} - \gamma_t$ and $f'(\tilde{\gamma}_{t+1}) - f'(\tilde{\gamma}_t)$
- Update $H_{t+1}$ using the Quasi-Newton formula.

If the change in $\gamma$ or the gradient is below a certain threshold, or if a maximum number of iterations is reached, stop the algorithm and return the approximated $\tilde{\gamma}^*$ as the result.

### B.2  Theoretical Guarantee

$$\min_{S \in M_n} ||S - S^0||_F^2 \implies \min_{v \in \mathbb{R}^n} ||v - v_0||^2$$
$$s.t. \ S \geq 0 \qquad\qquad s.t. \ v S_n^{-1} v \leq 1$$

The proof is as follows given by [37], where $S^*$ denotes the ground-truth of similarity matrix, $S^0$ denotes the initial similarity matrix calculated by incomplete data, and $\mathcal{T}$ denotes the feasible region $\{S \in M_n \mid S \geq 0, s_{ii} = 1, s_{ij} \in [-1, 1], \forall 1 \leq i \neq j \leq n\}$.

THEOREM B.1. $||S^* - \hat{S}||_F^2 \leq ||S^* - S^0||_F^2$. The equality holds if and only if $S^0 \in \mathcal{T}$, i.e., $S^0 = \hat{S}$.

PROOF. Let $M_n$ be the set of $n \times n$ symmetric matrices, equipped with an inner product that induces the Frobenius norm:

$$\langle X, Y \rangle = trace(X^\top Y), \text{ for any } X, Y \in M_n$$

Considering that $S^* \in \mathcal{T}$, we have:

$$||S^* - \hat{S}||_F^2$$
$$\leq ||S^* - \hat{S}||_F^2 - 2\langle S^* - \hat{S}, S^0 - \hat{S} \rangle$$
$$\leq ||S^* - \hat{S}||_F^2 + ||S^0 - \hat{S}||_F^2 - 2\langle S^* - \hat{S}, S^0 - \hat{S} \rangle$$
$$= ||(S^* - \hat{S}) - (S^0 - \hat{S})||_F^2$$
$$= ||S^* - S^0||_F^2$$

The first "$\leq$" holds due to Kolmogrov's criterion [17], which states that the projection of $S^0$ onto $\mathcal{T}$ is unique and characterized by:

$$\hat{S} \in \mathcal{T} \text{ and } \langle S - \hat{S}, S^0 - \hat{S} \rangle \leq 0 \text{ for all } S \in \mathcal{T}$$

The equality holds if and only if $\hat{S} = S^0$, i.e., $S^0 \in \mathcal{T}$.  □

### B.3  Positive Semi-definiteness

In the main text is a classical theorem based on Schur complement, which is proved as follows given by [21, 22, 30].

THEOREM B.2. Let $S_n \in \mathbb{R}^{n \times n}$ be a strictly positive definite matrix. Let $S_{n+1} = \begin{bmatrix} S_n & v \\ v^\top & 1 \end{bmatrix}$, where $v \in \mathbb{R}^n$. Then $S_{n+1}$ is positive semi-definite if and only if $v^\top S_n^{-1} v \leq 1$.

PROOF. Consider a matrix $S_{n+1} \in \mathbb{R}^{(n+1) \times (n+1)}$ partitioned as

$$S_{n+1} = \begin{bmatrix} S_n & v \\ v^\top & 1 \end{bmatrix}$$

where $S_n \in \mathbf{S}_{++}^n, v \in \mathbb{R}^n$. Due to $\det S_n \neq 0$, the Schur complement of $S_{pp}$ in $S$ is $D = 1 - v^\top S_{pp}^{-1} v$. Considering the minimization problem

$$\min \ u^\top S_{pp} u + 2\gamma^\top v^\top u + \gamma^\top 1 \gamma,$$

with variable $u \in \mathbb{R}^n, \gamma \in \mathbb{R}$. The optimal solution is $u^* = -S_{pp}^{-1} v \gamma$, we plugged it back into the minimization problem, and the minimization value is:

$$\inf_u \begin{bmatrix} u \\ \gamma \end{bmatrix}^\top \begin{bmatrix} S_{pp} & v \\ v^\top & 1 \end{bmatrix} \begin{bmatrix} u \\ \gamma \end{bmatrix}$$
$$= u^{*\top} S_{pp} u^* + 2\gamma^\top v^\top u^* + \gamma^\top 1 \gamma$$
$$= \gamma^\top v^\top S_{pp}^{-1} S_{pp} S_{pp}^{-1} v \gamma - 2\gamma^\top v^\top S_{pp}^{-1} v \gamma + \gamma^\top 1 \gamma$$
$$= \gamma^\top v^\top S_{pp}^{-1} v \gamma - 2\gamma^\top v^\top S_{pp}^{-1} v \gamma + \gamma^\top 1 \gamma$$
$$= -\gamma^\top v^\top S_{pp}^{-1} v \gamma + \gamma^\top 1 \gamma$$
$$= \gamma^\top (1 - v^\top S_{pp}^{-1} v) \gamma$$
$$= \gamma^\top D \gamma.$$

Then we obtain the equivalence of positive definiteness between $D$ and $S_{n+1}$:

$$D \geq 0 \Leftrightarrow \forall \gamma \in \mathbb{R}, \ \gamma^\top D \gamma \geq 0$$
$$\Leftrightarrow \forall \gamma \in \mathbb{R}, \ \inf_u \begin{bmatrix} u \\ \gamma \end{bmatrix}^\top S \begin{bmatrix} u \\ \gamma \end{bmatrix} \geq 0$$
$$\Leftrightarrow \forall u \in \mathbb{R}^n, \gamma \in \mathbb{R}, \ \begin{bmatrix} u \\ \gamma \end{bmatrix}^\top S \begin{bmatrix} u \\ \gamma \end{bmatrix} \geq 0$$
$$\Leftrightarrow S \geq 0,$$

which shows $S$ is positive semi-definite iff $v^\top S_{pp}^{-1} v \leq 1$.  □

### B.4  Convergence for CG-SVC

LEMMA B.3. If $A$ is PD, then $\lambda_{\min} \sqrt{n} \leq ||A||_F \leq \sqrt{n} \lambda_{\max}^2$; and

PROOF. By the property of $||A||_F$ [1],

$$||A||_F \leq \sqrt{r(A)} ||A||_2 \leq \sqrt{n} \lambda_{\max}^2$$

Moreover, by [2], $||A||_F \geq \sqrt{\sum_i \lambda_i^2} \geq \sqrt{n \lambda_{\min}^2} = \lambda_{\min} \sqrt{n}$.

□

---

[1] https://courses.cs.washington.edu/courses/cse521/17wi/521-lecture-8.pdf
[2] https://math.stackexchange.com/questions/620870/prove-that-the-square-sum-of-eigenvalues-is-no-more-than-the-frobenius-norm-for

**Table 4: Comparison of RMSE with $n = 2,000$ search candidates and $m = 1,000$ query items with various missing $\rho$ on different datasets. A smaller value denoted better performance.**

| $n$=2,000 $m$=1,000 | Missing Ratio | ZERO | MEAN | kNN | LR | RF | GROUSE | KFMC | DMC | CMC | SMC | QASMC | CQASMC |
|---|---|---|---|---|---|---|---|---|---|---|---|---|---|
| ImageNet | 0.2 | 4096.343 | 2.432 | 1.836 | 75.899 | 1.951 | 2.274 | 1.298 | 0.553 | 0.560 | 0.692 | 0.728 | 0.697 |
| | 0.5 | 3262.025 | 1.470 | 1.135 | 63.336 | 1.338 | 1.284 | 0.775 | 0.391 | 0.364 | 0.344 | 0.359 | 0.468 |
| | 0.8 | 2753.528 | 1.119 | 0.850 | 65.528 | 1.020 | 0.966 | 0.575 | 0.472 | 0.446 | 0.426 | 0.446 | 0.361 |
| MNIST | 0.2 | 1130.077 | 1.252 | 1.108 | 88.452 | 1.010 | 1.216 | 0.874 | 0.806 | 0.907 | 0.803 | 0.815 | 0.826 |
| | 0.5 | 2506.342 | 1.911 | 1.556 | 58.068 | 1.596 | 1.805 | 1.116 | 0.712 | 0.712 | 0.686 | 0.685 | 0.630 |
| | 0.8 | 3157.683 | 1.698 | 1.377 | 75.412 | 1.351 | 1.570 | 0.857 | 0.657 | 0.664 | 0.626 | 0.626 | 0.646 |
| CIFAR | 0.2 | 1951.028 | 1.776 | 1.539 | 108.977 | 1.433 | 1.702 | 1.103 | 0.742 | 0.800 | 0.713 | 0.717 | 0.726 |
| | 0.5 | 3691.816 | 2.257 | 1.804 | 77.694 | 1.779 | 2.092 | 1.065 | 0.572 | 0.578 | 0.533 | 0.533 | 0.522 |
| | 0.8 | 2634.238 | 1.445 | 1.182 | 5.604 | 1.289 | 1.228 | 0.627 | 0.444 | 0.420 | 0.400 | 0.400 | 0.418 |
| SIFT | 0.2 | 2219.104 | 2.011 | 1.568 | 87.813 | 1.513 | 1.799 | 1.915 | 0.596 | 0.589 | 0.536 | 0.541 | 0.543 |
| | 0.5 | 2341.015 | 2.029 | 1.538 | 92.612 | 1.525 | 1.845 | 1.980 | 0.593 | 0.601 | 0.557 | 0.557 | 0.577 |
| | 0.8 | 2451.514 | 2.419 | 1.518 | 89.890 | 1.472 | 1.784 | 1.893 | 0.582 | 0.614 | 0.578 | 0.568 | 0.583 |
| PROTEIN | 0.2 | 3149.420 | 1.897 | 1.458 | 88.809 | 1.419 | 1.786 | 0.994 | 0.756 | 0.756 | 0.756 | 0.754 | 0.761 |
| | 0.5 | 3436.216 | 1.945 | 1.591 | 80.079 | 1.574 | 1.801 | 0.977 | 0.618 | 0.621 | 0.583 | 0.583 | 0.587 |
| | 0.8 | 3112.226 | 1.796 | 1.473 | 61.915 | 1.445 | 1.676 | 0.972 | 0.635 | 0.630 | 0.597 | 0.596 | 0.582 |
| RCV1 | 0.2 | 3700.649 | 2.000 | 1.649 | 88.110 | 1.611 | 1.865 | 1.047 | 0.594 | 0.598 | 0.557 | 0.558 | 0.579 |
| | 0.5 | 3852.777 | 1.973 | 1.516 | 68.230 | 1.576 | 1.855 | 1.044 | 0.567 | 0.567 | 0.526 | 0.530 | 0.519 |
| | 0.8 | 3453.511 | 1.897 | 1.419 | 70.314 | 1.451 | 1.789 | 1.031 | 0.564 | 0.565 | 0.532 | 0.540 | 0.534 |

**Table 5: Comparison of RMSE with $n = 3,000$ search candidates and $m = 1,000$ query items with various missing $\rho$ on different datasets. A smaller value denoted better performance.**

| $n$=3,000 $m$=1,000 | Missing Ratio | ZERO | MEAN | kNN | LR | RF | GROUSE | KFMC | DMC | CMC | SMC | QASMC | CQASMC |
|---|---|---|---|---|---|---|---|---|---|---|---|---|---|
| ImageNet | 0.2 | 1899.345 | 1.625 | 1.355 | 141.023 | 1.252 | 1.562 | 0.973 | 0.750 | 0.812 | 0.723 | 0.726 | 0.696 |
| | 0.5 | 2628.687 | 1.274 | 1.012 | 65.396 | 1.166 | 1.117 | 0.652 | 0.463 | 0.431 | 0.417 | 0.416 | 0.498 |
| | 0.8 | 2415.008 | 1.262 | 0.981 | 64.555 | 1.113 | 1.090 | 0.617 | 0.484 | 0.460 | 0.441 | 0.442 | 0.440 |
| MNIST | 0.2 | 1999.841 | 1.453 | 1.180 | 137.948 | 1.100 | 1.385 | 0.844 | 0.756 | 0.821 | 0.803 | 0.731 | 0.728 |
| | 0.5 | 3256.074 | 1.994 | 1.642 | 60.128 | 1.642 | 1.867 | 1.033 | 0.624 | 0.628 | 0.686 | 0.588 | 0.563 |
| | 0.8 | 3502.127 | 2.145 | 1.775 | 81.767 | 1.688 | 1.992 | 1.105 | 0.579 | 0.587 | 0.626 | 0.541 | 0.546 |
| CIFAR | 0.2 | 2010.503 | 1.657 | 1.393 | 130.845 | 1.287 | 1.584 | 0.983 | 0.744 | 0.808 | 0.713 | 0.717 | 0.719 |
| | 0.5 | 2415.541 | 1.642 | 1.352 | 89.432 | 1.341 | 1.516 | 0.852 | 0.735 | 0.752 | 0.533 | 0.685 | 0.718 |
| | 0.8 | 2108.419 | 1.571 | 1.359 | 90.481 | 1.414 | 1.579 | 0.866 | 0.679 | 0.695 | 0.400 | 0.670 | 0.694 |
| SIFT | 0.2 | 2990.088 | 1.369 | 1.047 | 2.966 | 1.771 | 2.029 | 1.216 | 0.605 | 0.606 | 0.567 | 0.567 | 0.561 |
| | 0.5 | 2118.942 | 0.870 | 0.638 | 4.687 | 0.769 | 0.719 | 1.415 | 0.522 | 0.512 | 0.473 | 0.482 | 0.494 |
| | 0.8 | 2133.338 | 2.361 | 1.983 | 74.815 | 1.936 | 2.209 | 1.217 | 0.599 | 0.607 | 0.563 | 0.563 | 0.598 |
| PROTEIN | 0.2 | 3571.059 | 2.251 | 1.840 | 65.407 | 1.824 | 2.124 | 1.227 | 0.603 | 0.608 | 0.569 | 0.590 | 0.760 |
| | 0.5 | 3114.411 | 2.099 | 1.725 | 77.048 | 1.739 | 1.957 | 1.174 | 0.651 | 0.653 | 0.601 | 0.637 | 0.585 |
| | 0.8 | 3948.535 | 2.158 | 1.729 | 69.426 | 1.701 | 2.022 | 1.195 | 0.582 | 0.585 | 0.545 | 0.560 | 0.597 |
| RCV1 | 0.2 | 3701.289 | 2.277 | 1.953 | 85.739 | 1.858 | 2.141 | 1.274 | 0.589 | 0.590 | 0.552 | 0.552 | 0.538 |
| | 0.5 | 3333.193 | 2.607 | 2.077 | 60.643 | 2.067 | 2.410 | 1.387 | 0.624 | 0.628 | 0.593 | 0.589 | 0.569 |
| | 0.8 | 3486.254 | 2.282 | 1.891 | 58.856 | 1.847 | 2.123 | 1.209 | 0.600 | 0.602 | 0.566 | 0.562 | 0.563 |

**Table 6: Comparison of RMSE with $m = 4,000$ search candidates and $n = 1,000$ query items with various missing $\rho$ on different datasets. A smaller value denoted better performance.**

| $n$=4,000 $m$=1,000 | Missing Ratio | ZERO | MEAN | kNN | LR | RF | GROUSE | KFMC | DMC | CMC | SMC | QASMC | CQASMC |
|---|---|---|---|---|---|---|---|---|---|---|---|---|---|
| ImageNet | 0.2 | 2184.387 | 1.583 | 1.298 | 117.575 | 1.212 | 1.525 | 0.958 | 0.702 | 0.769 | 0.671 | 0.675 | 0.701 |
| | 0.5 | 1952.847 | 0.786 | 0.614 | 64.675 | 0.687 | 0.662 | 0.353 | 0.584 | 0.575 | 0.546 | 0.548 | 0.467 |
| | 0.8 | 2953.032 | 1.358 | 1.016 | 69.021 | 1.204 | 1.181 | 0.689 | 0.403 | 0.376 | 0.359 | 0.358 | 0.473 |
| MNIST | 0.2 | 3113.209 | 1.327 | 0.973 | 4.628 | 1.182 | 1.130 | 0.553 | 0.380 | 0.356 | 0.347 | 0.370 | 0.389 |
| | 0.5 | 2798.770 | 1.178 | 0.911 | 6.196 | 1.068 | 1.024 | 0.502 | 0.349 | 0.342 | 0.344 | 0.350 | 0.413 |
| | 0.8 | 2880.082 | 1.164 | 0.894 | 3.685 | 1.050 | 1.013 | 0.538 | 0.342 | 0.389 | 0.342 | 0.342 | 0.383 |
| CIFAR | 0.2 | 2162.085 | 1.643 | 1.366 | 141.266 | 1.259 | 1.581 | 0.989 | 0.715 | 0.785 | 0.672 | 0.687 | 0.694 |
| | 0.5 | 3359.900 | 2.144 | 1.694 | 60.944 | 1.717 | 2.014 | 1.110 | 0.605 | 0.608 | 0.570 | 0.570 | 0.579 |
| | 0.8 | 2175.928 | 1.120 | 0.898 | 4.819 | 1.000 | 0.946 | 0.551 | 0.538 | 0.516 | 0.494 | 0.495 | 0.490 |
| SIFT | 0.2 | 2901.842 | 2.512 | 1.854 | 69.319 | 1.931 | 2.415 | 1.681 | 0.689 | 0.692 | 0.641 | 0.645 | 0.650 |
| | 0.5 | 3143.413 | 2.142 | 1.785 | 78.210 | 1.942 | 2.084 | 1.519 | 0.575 | 0.578 | 0.551 | 0.551 | 0.566 |
| | 0.8 | 3099.105 | 2.515 | 1.742 | 67.894 | 1.988 | 2.131 | 1.451 | 0.390 | 0.379 | 0.356 | 0.357 | 0.408 |
| PROTEIN | 0.2 | 2589.518 | 1.895 | 1.413 | 76.141 | 1.415 | 1.859 | 1.052 | 0.689 | 0.690 | 0.665 | 0.691 | 0.650 |
| | 0.5 | 3262.297 | 1.991 | 1.553 | 75.549 | 1.574 | 1.860 | 1.007 | 0.641 | 0.645 | 0.628 | 0.607 | 0.566 |
| | 0.8 | 2141. 515 | 1.489 | 1.414 | 71.125 | 1.589 | 1.894 | 0.918 | 0.618 | 0.634 | 0.632 | 0.615 | 0.408 |
| RCV1 | 0.2 | 3624.475 | 2.181 | 1.793 | 87.477 | 1.775 | 2.029 | 1.203 | 0.592 | 0.597 | 0.556 | 0.556 | 0.563 |
| | 0.5 | 3056.585 | 1.952 | 1.587 | 81.825 | 1.561 | 1.797 | 0.989 | 0.654 | 0.661 | 0.620 | 0.618 | 0.613 |
| | 0.8 | 3056.783 | 1.956 | 1.641 | 79.417 | 1.598 | 1.799 | 1.000 | 0.642 | 0.653 | 0.605 | 0.606 | 0.609 |

**Table 7: Comparison of RMSE with $m = 5,000$ search candidates and $n = 1,000$ search candidates with various missing $\rho$ on different datasets. A smaller value denoted better performance.**

| n=5,000 m=1,000 | Missing Ratio | ZERO | MEAN | kNN | LR | RF | GROUSE | KFMC | DMC | CMC | SMC | QASMC | CQASMC |
|---|---|---|---|---|---|---|---|---|---|---|---|---|---|
| ImageNet | 0.2 | 1935.363 | 1.498 | 1.259 | 156.788 | 1.160 | 1.431 | 0.873 | 0.746 | 0.807 | 0.711 | 0.718 | 0.697 |
| | 0.5 | 3163.896 | 1.348 | 0.997 | 5.202 | 1.222 | 1.182 | 0.664 | 0.443 | 0.403 | 0.397 | 0.397 | 0.491 |
| | 0.8 | 1823.131 | 0.781 | 0.614 | 7.841 | 0.674 | 0.652 | 0.325 | 0.574 | 0.566 | 0.535 | 0.536 | 0.396 |
| MNIST | 0.2 | 2310.817 | 1.022 | 0.783 | 3.567 | 0.917 | 0.887 | 0.456 | 0.483 | 0.594 | 0.553 | 0.554 | 0.529 |
| | 0.5 | 2314.151 | 1.091 | 0.790 | 3.426 | 0.915 | 0.891 | 0.461 | 0.497 | 0.484 | 0.439 | 0.439 | 0.423 |
| | 0.8 | 2140.673 | 0.985 | 0.748 | 8.300 | 0.849 | 0.805 | 0.487 | 0.546 | 0.530 | 0.502 | 0.503 | 0.504 |
| CIFAR | 0.2 | 1926.079 | 1.781 | 1.531 | 114.301 | 1.361 | 1.700 | 1.098 | 0.745 | 0.804 | 0.720 | 0.720 | 0.726 |
| | 0.5 | 2014.142 | 0.945 | 0.741 | 65.019 | 0.835 | 0.806 | 0.489 | 0.515 | 0.520 | 0.473 | 0.474 | 0.475 |
| | 0.8 | 2425.479 | 0.957 | 0.724 | 67.901 | 0.831 | 0.819 | 0.479 | 0.492 | 0.489 | 0.452 | 0.454 | 0.485 |
| SIFT | 0.2 | 3253.020 | 2.138 | 1.874 | 60.949 | 1.835 | 2.032 | 1.266 | 0.642 | 0.644 | 0.601 | 0.616 | 0.637 |
| | 0.5 | 2134.877 | 1.341 | 1.431 | 65.145 | 1.031 | 0.849 | 1.654 | 0.512 | 0.564 | 0.564 | 0.567 | 0.564 |
| | 0.8 | 2883.282 | 1.110 | 0.840 | 64.416 | 1.002 | 0.963 | 1.560 | 0.411 | 0.383 | 0.355 | 0.356 | 0.362 |
| PROTEIN | 0.2 | 3223.142 | 2.156 | 1.642 | 54.142 | 2.641 | 2.433 | 1.141 | 0.652 | 0.651 | 0.631 | 0.664 | 0.694 |
| | 0.5 | 3440.121 | 2.454 | 2.037 | 87.979 | 2.018 | 2.287 | 1.271 | 0.617 | 0.621 | 0.579 | 0.581 | 0.602 |
| | 0.8 | 3413.145 | 2.514 | 2.314 | 79.314 | 1.909 | 2.151 | 1.159 | 0.619 | 0.631 | 0.589 | 0.589 | 0.622 |
| RCV1 | 0.2 | 3253.020 | 2.138 | 1.874 | 60.949 | 1.835 | 2.032 | 1.266 | 0.642 | 0.644 | 0.615 | 0.616 | 0.614 |
| | 0.5 | 3603.686 | 2.080 | 1.600 | 66.476 | 1.621 | 1.932 | 1.072 | 0.604 | 0.611 | 0.573 | 0.567 | 0.581 |
| | 0.8 | 2544.756 | 1.273 | 1.929 | 70.981 | 1.945 | 2.208 | 1.291 | 0.620 | 0.621 | 0.588 | 0.586 | 0.606 |

**Table 8: Comparison of Time Cost (seconds) with $n = 2,000$ search candidates and $m = 1,000$ query items with various missing $\rho$ on different datasets. A smaller value denoted better performance.**

| n=2,000 m=1,000 | Missing Ratio | DMC | CMC | SMC | QASMC | CQASMC |
|---|---|---|---|---|---|---|
| ImageNet | 0.2 | 97.605 | 214.391 | 200.530 | 76.146 | 27.015 |
| | 0.5 | 96.495 | 229.878 | 165.490 | 77.576 | 26.515 |
| | 0.8 | 100.027 | 282.974 | 134.880 | 81.349 | 26.484 |
| MNIST | 0.2 | 102.018 | 213.748 | 128.548 | 69.026 | 20.043 |
| | 0.5 | 96.366 | 227.495 | 128.651 | 68.566 | 22.054 |
| | 0.8 | 100.333 | 223.190 | 133.114 | 65.824 | 24.443 |
| CIFAR | 0.2 | 79.836 | 175.972 | 203.876 | 66.563 | 26.348 |
| | 0.5 | 74.198 | 174.743 | 141.386 | 74.008 | 27.850 |
| | 0.8 | 72.022 | 175.378 | 139.840 | 85.355 | 26.630 |
| SIFT | 0.2 | 69.174 | 100.043 | 209.686 | 88.548 | 26.278 |
| | 0.5 | 73.593 | 137.571 | 282.720 | 88.844 | 27.853 |
| | 0.8 | 75.296 | 141.379 | 305.000 | 91.662 | 28.783 |
| PROTEIN | 0.2 | 45.991 | 112.644 | 168.192 | 88.021 | 26.354 |
| | 0.5 | 37.136 | 95.537 | 198.990 | 89.065 | 27.055 |
| | 0.8 | 45.269 | 117.114 | 157.489 | 81.864 | 29.011 |
| RCV1 | 0.2 | 47.200 | 114.993 | 163.253 | 79.880 | 26.435 |
| | 0.5 | 47.465 | 121.871 | 157.749 | 80.229 | 26.932 |
| | 0.8 | 54.696 | 126.589 | 158.662 | 81.353 | 27.260 |

**Table 9: Comparison of Time Cost (seconds) with $n = 3,000$ search candidates and $m = 1,000$ query items with various missing $\rho$ on different datasets. A smaller value denoted better performance.**

| n=3,000 m=1,000 | Missing Ratio | DMC | CMC | SMC | QASMC | CQASMC |
|---|---|---|---|---|---|---|
| ImageNet | 0.2 | 97.605 | 214.391 | 200.530 | 76.146 | 27.015 |
| | 0.5 | 96.495 | 229.878 | 165.490 | 77.576 | 26.515 |
| | 0.8 | 100.027 | 282.974 | 134.880 | 81.349 | 26.484 |
| MNIST | 0.2 | 102.018 | 213.748 | 128.548 | 69.026 | 20.043 |
| | 0.5 | 96.366 | 227.495 | 128.651 | 68.566 | 22.054 |
| | 0.8 | 100.333 | 223.190 | 133.114 | 65.824 | 24.443 |
| CIFAR | 0.2 | 79.836 | 175.972 | 203.876 | 66.563 | 26.348 |
| | 0.5 | 74.198 | 174.743 | 141.386 | 74.008 | 27.850 |
| | 0.8 | 72.022 | 175.378 | 139.840 | 85.355 | 26.630 |
| SIFT | 0.2 | 69.174 | 100.043 | 209.686 | 88.548 | 26.278 |
| | 0.5 | 73.593 | 137.571 | 282.720 | 88.844 | 27.853 |
| | 0.8 | 75.296 | 141.379 | 305.000 | 91.662 | 28.783 |
| PROTEIN | 0.2 | 45.991 | 112.644 | 168.192 | 88.021 | 26.354 |
| | 0.5 | 37.136 | 95.537 | 198.990 | 89.065 | 27.055 |
| | 0.8 | 45.269 | 117.114 | 157.489 | 81.864 | 29.011 |
| RCV1 | 0.2 | 47.200 | 114.993 | 163.253 | 79.880 | 26.435 |
| | 0.5 | 47.465 | 121.871 | 157.749 | 80.229 | 26.932 |
| | 0.8 | 54.696 | 126.589 | 158.662 | 81.353 | 27.260 |

LEMMA B.4. *If $A$ is PD, then $||x||_A := x^\top A x \leq \lambda_{\max}(A) x^\top x$ and $||x||_A := x^\top A x \geq \lambda_{\min}(A) x^\top x$*

PROOF. Note that
$x^\top A x = \langle x, Ax \rangle = \langle x, U^\top \Sigma U x \rangle = \langle U^\top x, \Sigma U x \rangle = \sum_i \lambda_i y_i^2$, where $y = Ux$, and $||y|| = ||x||$.

Thus, taking the maximum and min-mum $\lambda$, we obtain our result.
□

LEMMA B.5. *Assume $|| \left( S_{\mathrm{pp}}^{-1} - S_{\mathrm{pp}}^{-1}{}' \right) ||_F \leq \epsilon_1$, $||\tilde{\gamma} - \tilde{\gamma}'||_F \leq \epsilon_2$, we have*

$$|| \left( I + \tilde{\gamma}^* S_{\mathrm{pp}}^{-1} \right) v^0 - \left( I + \tilde{\gamma}' S_{\mathrm{pp}}^{-1}{}' \right) v^0 || \leq \left( \tilde{\gamma}^* \epsilon_1 + \sqrt{n} \lambda_{\max}^2 \left( S_{\mathrm{pp}}^{-1} \right) \epsilon_2 \right) n$$

*where the dimension of $v_0$ is n.*

PROOF.

$$\left\| \left( I - 2\tilde{\gamma}^* S_{\mathrm{pp}}^{-1} \right) v^0 - \left( I - 2\tilde{\gamma}' S_{\mathrm{pp}}^{-1} \right) v^0 \right\|$$

$$\leq 2 \left\| \tilde{\gamma}^* S_{\mathrm{pp}}^{-1} - \tilde{\gamma}' S_{\mathrm{pp}}^{-1} \right\|_F \| v^0 \|$$

$$\leq 2 \left( \left\| \tilde{\gamma}^* \left( S_{\mathrm{pp}}^{-1} - S_{\mathrm{pp}}^{-1} \right) \right\|_F + \left\| S_{\mathrm{pp}}^{-1} (\tilde{\gamma} - \tilde{\gamma}') \right\|_F \right) \| v^0 \|$$

$$\overset{(a)}{\leq} 2 \left( \tilde{\gamma}^* \epsilon_1 + \left\| S_{\mathrm{pp}}^{-1} \right\| \epsilon_2 \right) n$$

$$\overset{(b)}{\leq} 2 \left( \tilde{\gamma}^* \epsilon_1 + \sqrt{n} \lambda_{\max}^2 \left( S_{\mathrm{pp}}^{-1} \right) \epsilon_2 \right) n$$

□

(a) is due to $|| \left( S_{\mathrm{pp}}^{-1} - S_{\mathrm{pp}}^{-1}{}' \right) ||_F \leq \epsilon_1$, $||\tilde{\gamma} - \tilde{\gamma}'||_F \leq \epsilon_2$.
(b) is due to Lemma B.3.

**Table 10: Comparison of Time Cost (seconds) with $n = 4,000$ search candidates and $m = 1,000$ query items with various missing $\rho$ on different datasets. A smaller value denoted better performance.**

| $n=4,000$ $m=1,000$ | Missing Ratio | DMC | CMC | SMC | QASMC | CQASMC |
|---|---|---|---|---|---|---|
| **ImageNet** | 0.2 | 96.291 | 205.023 | 200.560 | 81.420 | 27.115 |
| | 0.5 | 94.474 | 272.922 | 137.070 | 75.697 | 26.329 |
| | 0.8 | 91.814 | 219.453 | 161.170 | 76.795 | 27.014 |
| **MNIST** | 0.2 | 95.210 | 221.877 | 165.767 | 70.480 | 26.419 |
| | 0.5 | 97.573 | 228.800 | 155.451 | 73.028 | 27.960 |
| | 0.8 | 93.691 | 240.372 | 158.634 | 77.608 | 27.632 |
| **CIFAR** | 0.2 | 80.883 | 144.731 | 196.658 | 85.878 | 27.028 |
| | 0.5 | 88.882 | 225.155 | 250.973 | 77.199 | 28.632 |
| | 0.8 | 89.138 | 233.421 | 281.737 | 85.695 | 22.478 |
| **SIFT** | 0.2 | 74.145 | 99.589 | 202.988 | 80.939 | 29.270 |
| | 0.5 | 79.708 | 199.163 | 246.371 | 85.649 | 22.169 |
| | 0.8 | 75.280 | 183.807 | 260.078 | 93.261 | 26.467 |
| **PROTEIN** | 0.2 | 46.742 | 110.897 | 167.928 | 82.548 | 27.403 |
| | 0.5 | 41.758 | 94.208 | 196.647 | 82.890 | 26.630 |
| | 0.8 | 45.296 | 108.610 | 158.170 | 87.747 | 26.511 |
| **RCV1** | 0.2 | 48.606 | 113.606 | 154.619 | 76.131 | 26.354 |
| | 0.5 | 46.940 | 107.666 | 165.954 | 78.400 | 26.277 |
| | 0.8 | 48.032 | 113.999 | 155.026 | 76.713 | 26.189 |

**LEMMA B.6.** *if $f$ is strongly convex and $\nabla^2 f(x)$ is Lipschitz continuous, we have*

$$\left\| x_{k+1} - x^\star \right\|_2 \leq c_k \left\| x_k - x^\star \right\|_2$$

*where $c_k$ is a constant ,and $0 < c_k < 1$.*

PROOF. This can be found in page 9 http://www.seas.ucla.edu/~vandenbe/236C/lectures/qnewton.pdf. □

**LEMMA B.7.** *Let $x_k$ be the $k$-th iteration of the CG method with $X_0$. For PD matrix $S_{pp}$,*

$$\left\| x^* - x_k \right\|_A \leq 2 \left( \frac{\sqrt{\kappa(S_{pp})} - 1}{\sqrt{\kappa(S_{pp})} + 1} \right)^k \left\| x^* - x_0 \right\|_A$$

*where $\kappa(S_{pp}) = \lambda_{\max}(S_{pp})/\lambda_{\min}((S_{pp}))$*

PROOF. Proof is on theorem 3.5 in https://www.math.uci.edu/~chenlong/226/CG.pdf. □

**LEMMA B.8.** *If $A$ is PD, then*
$$\lambda_{\min}(S_{pp}) \sum_i \left\| x^* - x_i^k \right\| \leq 2 \left( \frac{\sqrt{\kappa(S_{pp})} - 1}{\sqrt{\kappa(S_{pp})} + 1} \right)^k \lambda_{\max}(S_{pp}) \sum_i \left\| x^* - x_i^k \right\|.$$

PROOF. From Lemma B.7 and Lemma B.4
$$\lambda_{\min}(S_{pp}) \sum_i \left\| x^* - x_i^k \right\|$$

$$\leq \sum_i \left\| x^* - x_i^k \right\|_A \leq 2 \left( \frac{\sqrt{\kappa(S_{pp})} - 1}{\sqrt{\kappa(S_{pp})} + 1} \right)^k \sum_i \left\| x^* - x_i^0 \right\|_A$$

$$\leq 2 \left( \frac{\sqrt{\kappa(S_{pp})} - 1}{\sqrt{\kappa(S_{pp})} + 1} \right)^k \lambda_{\max}(S_{pp}) \sum_i \left\| x^* - x_i^k \right\|.$$

□

**LEMMA B.9.** $\left( I + \tilde{\gamma}^* S_{pp}^{-1} \right) v^0 \leq \left( \frac{c}{\lambda_{\min}(I + 2\tilde{\gamma}^* S_{pp}^{-1})} \right)^3 n$, *where* $0 < c < \lambda_{\min}$.

$$\left\| \left( I - 2\tilde{\gamma}^* S_{pp}^{-1} \right) v^0 - \left( I + 2\tilde{\gamma}^* S_{pp}^{-1} \right)^{-1} v^0 \right\|$$

$$\leq \left\| \left( I - 2\tilde{\gamma}^* S_{pp}^{-1} \right) v^0 - \left( I + 2\tilde{\gamma}^* S_{pp}^{-1} \right)^{-1} \right\| \| v^0 \|$$

$$\leq \left( \frac{c}{\lambda_{\min}(I + 2\tilde{\gamma}^* S_{pp}^{-1})} \right)^3 n$$

The last inequality is based on the result from [52].

**THEOREM B.10.** *When running conjugate gradient with $k_1$ step, and Quasi-Netwon method with $k_2$ step, QABSVC has*

$$\left\| v^* - \left( I + 2\tilde{\gamma}' S_{pp}^{-1} \prime \right)^{-1} v^0 \right\| \leq O(c_0^{k_1}) + O(c_1^{k_2})$$

*where $0 < c_0 < 1$, and $0 < c_1 < 1$.*

PROOF. Noting that $v^* = \left( I + 2\gamma^* S_{pp}^{-1} \right)^{-1} v^0$ by the KKT condition, where $\gamma^*$ satisfying $\left( \left( I + 2\tilde{\gamma} S_{pp}^{-1} \right)^{-1} v^0 \right)^\top S_{pp}^{-1} \left( I + 2\tilde{\gamma} S_{pp}^{-1} \right)^{-1} v^0 = I$. By directly com-binding the lemma B.5,

$$\left\| v^* - \left( I + \tilde{\gamma}' S_{pp}^{-1} \prime \right) v^0 \right\| \leq \left( \left( \tilde{\gamma}^* \epsilon_1 + \sqrt{n} \lambda_{\max}^2 \left( S_{pp}^{-1} \right) \epsilon_2 \right) n \right.$$

Then, by using Lemma B.8 Lemma B.6,

$$\lambda_{\min}(S_{pp}) \epsilon_1 \leq 2 \left( \frac{\sqrt{\kappa(S_{pp})} - 1}{\sqrt{\kappa(S_{pp})} + 1} \right)^{k_1} \lambda_{\max}(S_{pp}) \left\| S_{pp}^{-1} - S_0 \right\|.$$

$$\epsilon_2 \leq 2c_1^{k_2} \| \gamma_0 - \gamma^* \|$$

Then, we have:

$$\left\| v^* - \left( I + 2\tilde{\gamma}' S_{pp}^{-1} \prime \right)^{-1} v^0 \right\|$$
$$\leq O(c_0^{k_1}) + O(c_1^{k_2})$$

□