# OpenReview forum: "A Matrix Calibration Method for Similarity Matrix with Incomplete Query"
_ACM.org/TheWebConf/2024/Conference — TheWebConf24 Oral_

### Official Review · Reviewer_Rasf · 2023-11-22

**Novelty:** 6
**Technical Quality:** 5

**Review:**

The paper proposes a novel similarity matrix calibration method. The authors propose a parallel matrix calibration method to estimate the similarity matrix to approximate the unknown fully observed ground-truth similarity matrix. They also discover its factored form, which bypasses the computation of singular values and allows fast optimization by a general optimization algorithm. Stable recovery and convergence are guaranteed. Extensive similarity matrix calibration experiments on the real-world dataset demonstrated that the proposed method obtains superior performance while being the fastest compared to baseline methods.

The idea presented in the paper is interesting. The paper falls within the WWW 2024 call for papers, and the similarity search problem justifies the need for such an approach. The paper is an interesting piece of work to read. The novelty is multifold. Two different algorithms are proposed: first, the authors introduce the Basic Similarity Vector Calibration (BSVC) method to solve the similarity search problem with incomplete queries. Then they further reduce the computational complexity with Conjugate Gradient and Quasi-Newton-based Approximated BSVC (CQABSVC) methods to find the approximated solutions. The novelty of the paper is represented by the two algorithms and their theoretical justification. Moreover, the authors propose an experimental evaluation of the performance of the two new algorithms on five public datasets. The new proposed approaches significantly advance the state of the art. From an experimental point of view, the implementation of the algorithms (MATLAB) is not available.

From an experimental point of view, while Table 1 looks sound, some of the details in Table 2 are unclear. First, it is not clear what are the results from n = 1,000 and n = 5,000. Assuming n = 1,000 is the table on the left, why for some datasets (CIFAR / SIFT / PROTEIN) computing for n = 1,000 costs more than computing for n = 5,000 (SMC, QASMC)? Please explain this behavior.

Overall, although the paper significantly falls outside my main area of expertise, I found it an interesting piece of work to read, and I think it reaches the bar for WWW 2024.

**Questions:**

1) Could you please explain the behavior shown in Table 2 for some datasets? Why do some methods present lower time costs for n = 5,000? IMHO it is counterintuitive.

The authors answer the question above during rebuttal.

**Ethics Review Description:**

does not apply

**Reviewer Confidence:**

3: The reviewer is confident but not certain that the evaluation is correct

**Scope:**

3: The work is somewhat relevant to the Web and to the track, and is of narrow interest to a sub-community

---

### Official Review · Reviewer_Bfra · 2023-11-23

**Novelty:** 5
**Technical Quality:** 5

**Review:**

Advantage：
1.Quality: This paper proposes a new similarity matrix calibration method and efficient algorithm to address incomplete observations in similarity search problems. The quality of the paper is high.

2.Clarity: The discourse of the paper is clear and clear, and the content of each part is naturally connected and easy to understand. For the proposed algorithm, the paper provides pseudo code with detailed steps and explanations, making it easy for readers to understand and implement.

3.Originality: This paper proposes a novel matrix calibration method that use positive semi-definiteness (PSD) property to estimate the similarity matrix from the incomplete data. This method has not been reported in existing literature, therefore it has high originality.

4.Significance: The method proposed in this paper is of great significance for solving incomplete observations of similarity search problems. In real life, incomplete observations are a common phenomenon, so this method has broad application prospects.

Disadvantage:
1.For certain specific situations, this method may not achieve the best results. The experimental results in the paper also indicate that the method did not achieve optimal results in all cases.

2.The paper conducted experiments on five datasets, but only one text dataset was included. This cannot fully demonstrate the effectiveness of this method in different fields and datasets with higher dimensions.

**Questions:**

1.The article mentions the use of cosine similarity as a measure. However, in other contexts, other similarity measures such as Euclidean distance, Jaccard index, etc. may be more appropriate. Can the methods in the article be extended to cover these other similarity measures?

2.The article mentions various application scenarios for matrix calibration problems, but does not delve into the specific implementation details and performance evaluation in these application scenarios. If more information can be provided on the specific implementation and performance evaluation in these application scenarios, it will help readers better understand how to use matrix calibration methods in practical problems.

3.If the experimental results on more text datasets of the methods in the article can be provided, it will greatly improve the persuasiveness of the article.

**Reviewer Confidence:**

3: The reviewer is confident but not certain that the evaluation is correct

**Scope:**

3: The work is somewhat relevant to the Web and to the track, and is of narrow interest to a sub-community

---

### Official Review · Reviewer_jhKk · 2023-11-29

**Novelty:** 5
**Technical Quality:** 3

**Review:**

To solve the problem of incomplete data when calculate the similarity matrix, this paper introduces a new algorithm based on the positive semi-definiteness (PSD) property of the similarity matrix, where the similarity matrix is estimated by approximating the unknown fully observed ground-truth similarity matrix iteratively. Overall, the motivation and formula derivation process are clear. However, the results and results analysis are confusing and incorrect sometimes.
Cons:
(1)The caption of Table 1 is wrong. “n=1000 query items” should be “n=1000 complete search samples” as described in the former sections.
(2)In section 6.5, the conclusion that “ Overall, the RMSE decreased with the increase of missing ratio rho” is wrong by comparing the fig1-fi3. For example, taking the SIFT as example, we can not draw this conclusion. Besides, to check the changes of RMSE of different rho, it is better to merge the figure1-3 together. Currently, it is difficult to draw a right conclusion from these three separate figures.
(3)The conclusion that “RMSE was not much changed with a fixed missing ratio on a specific rho in most cases.” is also wrong in section 6.5. It is apparent that RMSE changes a lot for specific tol1 for CG and tol2 for QN in the MINIST, CIFAR, PROTEIN and RCV1 datasets. Such situation is apparent for different rho.
(4)To verify the goodness of the matrix calibration method, it is better to conduct experiments at a specific retrieval task using the similarity matrix. In this way, the superiority of the proposed method can be better validated.

**Questions:**

(1)In the similarity matrix initialization phase, what is the meaning of “common features that are observed in both x_i and x_j.” Can you give an example?
(2)Can you explain more details about the adopted evaluation metric? Has this metric been used by other baseline methods? Does this metric is suitable for the Missing Value Imputation Methods?  As the authors report, the RMSE of Missing Value Imputation Methods is really bigger than the proposed method. So I am curious if the adopted metric is suitable for these baseline methods.
(3)The authors should concern the conclusion statements concluded from the tables and figures.

**Ethics Review Description:**

No ethical issue.

**Reviewer Confidence:**

3: The reviewer is confident but not certain that the evaluation is correct

**Scope:**

3: The work is somewhat relevant to the Web and to the track, and is of narrow interest to a sub-community

---

### Official Review · Reviewer_Y2Tp · 2023-11-29

**Novelty:** 5
**Technical Quality:** 6

**Review:**

The paper describes a fast matrix calibration method when a full similarity matrix is not available due to storage/transmission issues. The paper details the theoretical foundation of the proposed approach, describes algorithm and overheads and demonstrates the improved performance in relation to 2 baselines.

Pros:
- (as far as I can tell) technically interesting, novel approach

Cons:
- The graphs and figures in Section 6 are so small that they're basically unreadable in the printed form of the paper.
- There are a number of spelling/grammar issues in the paper and the English is often difficult to follow...
- The two motivating scenarios regarding missing entries in the similarity matrix outline in the first paragraph of the introduction both aren't fully compelling; what do the authors mean exactly by "being measures by the incomplete data samples"? What is the cause for features being unknown? A more concrete example would be very beneficial here. Also, with error-correcting codes and redundant storage systems, it is not clear to be my storage/transmission errors should be a common occurrence. This is especially true as Section 2.2.  suggests that a somewhat large fraction of the data should be missing for the proposed method to be applicable.
- The use of the calibrated similarity metric in the context of similarity search isn't fully compelling either; to the best of my knowledge, nearly all ANN techniques use relatively simple similarity measures over vectors of values (or a model to approximate the similarity function), but not explicitly materialized matrices.
- While similarity estimation is relevant to several different types of applications that are relevant to the Web Conference, I am less sure that the technical contributions of this paper would be a good match with the conference audience.

**Questions:**

- Are there any other application scenarios (other than data being lost in storage/transit) for which the proposed techniques are relevant?
- It would be important to understand what other similarity measures the proposed method can reconstruct (Section 3.2. only gives pointers to [35,48]) and I would like the authors to describe this in more detail (and also, if and how this would affect accuracy).
- What do the authors mean by "the assumption of data samples" (Section 1)?

**Reviewer Confidence:**

1: The reviewer's evaluation is an educated guess

**Scope:**

2: The connection to the Web is incidental, e.g., use of Web data or API

---

### Decision · Program_Chairs · 2024-01-22

**Decision:**

Accept (Oral)

**Comment:**

This is the meta-review. The paper proposes a novel similarity matrix calibration method to estimate and approximate the unknown fully observed similarity matrix with incomplete query.

 During the discussion phase, the authors provided thorough information. Additional experimental results have been added. Some major and common concerns of the original reviews have been clarified clearly, such as adding various similarity metrics, adding one additional text dataset, and better explanations on data missing scenarios.

 Pros:
 + The idea is interesting and it studies an important and common problem in real applications.
 + The proposed approach has technical quality and novelty, which uses the positive semi-definiteness (PSD) property to estimate the similarity matrix from the incomplete data.
 + Extensive experiments have been performed on 5+1 dataset, and the comparative results verify that the model is effective (in most cases) and efficient.

 Cons:
 - The writing can be improved, which the author has promised to be improved in the revised version.
 - I'm somewhat worrying about whether the additional experimental results and major explanations can be fully addressed in the revised paper. But such supplement information is necessary and helpful. Careful re-organization and improved representation of the content is needed.

 Given the above evaluation, and since no paper is perfect, I think the paper is ready for publication, if the authors do add these improvements during the discussions into the final version.